**RESEARCH**

# Strand asymmetry influences mismatch resolution during a single-strand annealing

Victoria O. Pokusaeva[1,2], Aránzazu Rosado Diez[1,3], Lorena Espinar[1], Albert Torelló Pérez[1] and Guillaume J. Filion[1,4,5*]

*Correspondence: guillaume.filion@gmail.com
[5] Present Address: Department Biological Sciences, University of Toronto Scarborough, Toronto, Canada
Full list of author information is available at the end of the article

## Abstract

**Background:** Biases of DNA repair can shape the nucleotide landscape of genomes at evolutionary timescales. The molecular mechanisms of those biases are still poorly understood because it is difficult to isolate the contributions of DNA repair from those of DNA damage.

**Results:** Here, we develop a genome-wide assay whereby the same DNA lesion is repaired in different genomic contexts. We insert thousands of barcoded transposons carrying a reporter of DNA mismatch repair in the genome of mouse embryonic stem cells. Upon inducing a double-strand break between tandem repeats, a mismatch is generated if the break is repaired through single-strand annealing. The resolution of the mismatch showed a 60–80% bias in favor of the strand with the longest 3′ flap. The location of the lesion in the genome and the type of mismatch had little influence on the bias. Instead, we observe a complete reversal of the bias when the longest 3′ flap is moved to the opposite strand by changing the position of the double-strand break in the reporter.

**Conclusions:** These results suggest that the processing of the double-strand break has a major influence on the repair of mismatches during a single-strand annealing.

**Keywords:** Mismatch repair, Single-strand annealing, Chromatin, Genome-wide technologies, Mouse embryonic stem cells

## Background

The genome of every organism is the result of a mutation–selection process that unfolds since the origins of life. Mutations have a dual role in this process: on the one hand, they generate the diversity for selection to act upon; on the other hand, they drive evolution through non-selective forces [1]. Non-selective forces are changes that drive a genome away from its current state without affecting the fitness of the organism. For instance, small asymmetries in mutations can accumulate over evolutionary timescales so as to form sequence patterns in a genome [2, 3].

At least two features of mammalian genomes are shaped by non-selective forces: the depletion of CpG dinucleotides [4] and the 10 bp periodicity of ApA dinucleotides [5]. The first is due to increased C to T mutations when C is methylated, which takes place only within CpG dinucleotides [6–8]. The second is due to increased damage on nucleotides facing outward the nucleosome, where ApA is the least exposed dinucleotide [1, 9, 10].

The mechanisms that underlie mutational biases in mammals are otherwise poorly understood. For instance, one of the most enigmatic features of vertebrate genomes is that they are organized in megabase-scale domains called isochores, where the GC content is relatively uniform [11]. However, the average GC content varies from 30 to 70% between isochores. Several lines of evidence suggest that the pattern may emerge from an asymmetry in meiotic recombination known as biased gene conversion [12, 13]. The theory postulates that mismatched heteroduplexes are repaired in favor of G/C alleles over A/T alleles, with the consequence that the GC content increases at recombination hotspots [14].

Initial estimates in mammalian genomes suggested that mismatch repair is indeed biased toward G and C nucleotides [15], but those estimates were obtained on circular unintegrated plasmids. In more realistic conditions where heteroduplexes are integrated and repaired in the genome, the bias disappeared, except for the G:T mismatch, handled by a specific repair pathway [16]. Also, it was later shown that in mice, GC-biased gene conversion is restricted to non-crossover events with a single mismatch [17]. It is thus doubtful that any nucleotide is intrinsically favored by the mismatch repair system. Instead, it appears that there exists a hierarchy of factors influencing repair biases, but which take precedence over the others is largely unknown.

Recent insight into this question came from cancer genome sequencing [18, 19]. In particular, this made it possible to show that the mismatch repair system in healthy cells is more accurate at some loci than others. For instance, mismatches in late-replicating regions are repaired less efficiently [20], a feature that seems to be shared among eukaryotes [21]. It is presently unknown whether the chromatin context can bias the mismatch repair toward one nucleotide or another, mostly because it is difficult to tease apart the contributions of DNA damage and DNA repair to mutation patterns.

In sum, the fact that DNA repair is context-dependent suggests that it may have a large influence on the local nucleotide composition of a genome. However, the importance of chromatin compared to the molecular features of the lesion is an open question. More generally, it has been so far impossible to separate the biases due to damage from the biases due to repair in the context of chromatin, mostly for lack of a technology to engineer and track mismatches genome-wide.

Here, we set out to measure such biases in the chromatin of wild-type mouse embryonic stem cells (ES cells). We develop an assay where a mismatch is produced in the genome as a byproduct of the single-strand annealing pathway (SSA). Using reporters integrated at tens of thousands of locations, we pit nucleotides against each other. This allows us to directly test the hypothesis that the mismatch repair pathway favors G/C alleles over A/T alleles in different genomic contexts.

## Results

### A TRIP assay to measure mismatch repair biases in SSA

TRIP (Thousands of Reporters Integrated in Parallel) is a shotgun technique to assay the influence of the chromatin context on a phenomenon of interest [22–24]. The principle is to insert reporters at different locations of the genome and to measure a readout in bulk using DNA barcodes (Fig. 1a). The experiment typically consists of two phases: in the first, transposons are inserted in a cell pool and the barcodes are mapped to generate a lookup table; in the second phase, the phenomenon of interest is measured for all the barcodes simultaneously and the outcome is demultiplexed using the lookup table.

In this study, we developed a TRIP assay to measure biases of mismatch resolution in the DNA. The reporter construct consists of two nearly identical 152 bp sequences separated by a restriction site for the meganuclease I-SceI (Fig. 1b). The 152-bp sequence is a middle segment of the coding sequence of GFP, so we refer to it as "F segment" throughout. The F segments differ by one nucleotide located at the center, so annealing two strands from different F segments creates a 152-bp heteroduplex with a central mismatch. The assay is initiated by expressing I-SceI, which cleaves the integrated reporters (Fig. 1c). The double-strand breaks are repaired by either non-homologous end joining (NHEJ) or single-strand annealing (SSA). In the first case, DNA ends are blunted and ligated so the final product consists of two F segments with distinct alleles, flanking the scarred I-SceI site. In the second case, 5' DNA ends are resected and the two strands anneal to each other, forming a mismatched duplex that is eventually repaired [25]. The final product contains only one F segment with only one of the two original alleles.

The outcome of mismatch repair is revealed by sequencing the reporters that have only one F segment. The barcode flanking the sequence allows us to know the location of the reporter in the genome of mouse ES cells thanks to the lookup table.

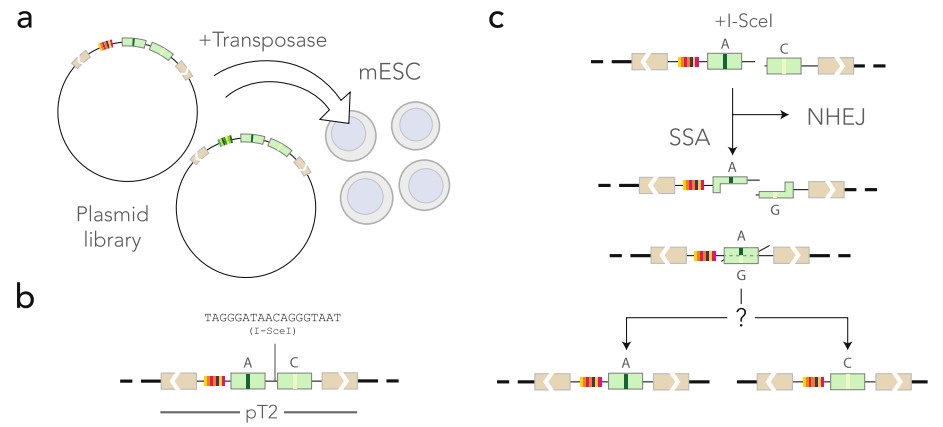

**Fig. 1** Experimental approach. **a** Mouse ES cells in culture are co-transfected with a barcoded reporter library and with the Sleeping Beauty 100X transposase. **b** The reporters in the pT2 transposon backbone are integrated at random in the mouse genome. **c** Mismatches occur during the repair of a double-strand break induced by the transient expression of I-SceI. If the double-strand break is repaired through non-homologous end joining (NHEJ), no mismatch is formed. If it is repaired by single-strand annealing (SSA), a mismatch is formed and repaired in favor of one of the two nucleotides. Sequencing the construct reveals the outcome of DNA repair at different locations identified through the barcode

## The integrated reporters cover the mouse genome

We designed four similar constructs where SSA produces heteroduplexes with A:G, T:G, A:C, and T:C mismatches. In those constructs, the A and T alleles are on the left F segment in the orientation of Fig. 1b, while G and C alleles are on the right one. This means that during heteroduplex formation, A/T nucleotides are always of the top strand in the orientation of Fig. 1c and G/C nucleotides on the bottom strand. We therefore included a strand-swap control of the T:G mismatch, referred to as G:T, where G is on the top strand and T is on the bottom strand. For concreteness, we will refer to the strands as "top" and "bottom" in what follows, always assuming that the reference orientation is that of Fig. 1b.

The five constructs were barcoded using random 20-mers so that each integrated reporter contains a different barcode (see the "Methods" section). The barcoded TRIP reporter libraries were inserted by Sleeping Beauty transposition in the genome of E14 mouse ES cells [26]. After 2 weeks of growth, the reporters were mapped by inverse PCR (see the "Methods" section). For each construct, we mapped between 9 and 48 thousand reporters, with 107 thousand known reporter locations in total (see Table 1). The substantial variations in the number of recorded events are due to batch effects, rather than differences in the induction of the double-strand break or in the efficiency of the DNA repair system. For the T:C construct in particular, many sequencing reads were lost due to contaminations of the PCR products.

At chromosomal scale, the transposons were found everywhere except on the Y chromosome (Fig. 2a). E14 ES cells are male, but the repetitive nature of the Y chromosome makes it impossible to map the reporters unambiguously. The insertion rate on the X chromosome was approximately half the value observed on the autosomes, in line with the expectation for a male cell line (Fig. 2b). This shows that the reporters were distributed evenly throughout the mappable genome.

Overall, the mapped reporters were enriched in transcribed genes, with a 35% excess over random (Fig. 2c). Genes are typically more mappable than the rest of the genome, but mapped reporters were not enriched in silent genes, suggesting that ongoing transcription facilitates the insertion of the transposon. Meanwhile, reporters were depleted

**Table 1** Mapping and repair statistics. *Construct*: code of the mismatch produced during SSA, with the convention that the left nucleotide is proximal to the barcode (for instance, the construct in Fig. 1b is A:G). *Mapped*: number of barcodes unambiguously mapped in the mouse genome (sum of two TRIP pools). *In repeats*: number of barcodes mapped in repeated sequences. *Repair events*: number of barcodes for which the outcome of mismatch resolution was measured (technical replicates are counted separately). *Mapped events*: subset of the repair events where the barcode is mapped unambiguously

| Construct | Mapped | In repeats | Repair events | Mapped events |
|---|---|---|---|---|
| A:G | 15,601 | 6,346 | 28,115 | 8,738 |
| T:G | 9,451 | 3,619 | 188,575 | 24,856 |
| A:C | 12,445 | 4,963 | 32,872 | 9,138 |
| T:C | 21,448 | 8,363 | 4,189 | 965 |
| G:T | 48,627 | 19,575 | 75,775 | 39,227 |
| Total | 107,572 | 42,866 | 329,526 | 82,924 |

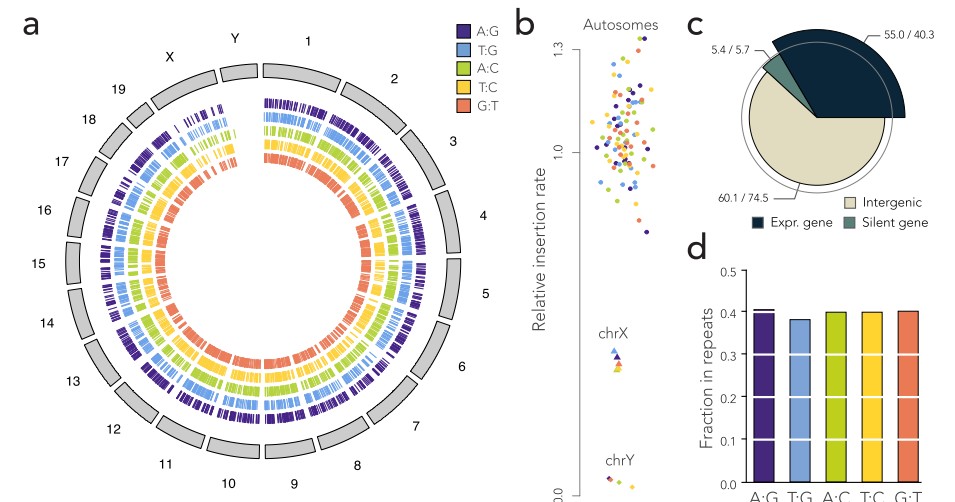

**Fig. 2** Mapping of the reporters. **a** Overview of the insertions. For each construct, 1000 insertions were drawn at random and plotted on a circular representation of the mouse genome. Each tick mark represents a mapped insertion. **b** Dot plot of the relative insertion rate per chromosome. For each chromosome and each construct, the insertion rate was computed as the number of insertions per bp, normalized by the expected number of insertions under the uniform model. **c** Insertions relative to genes. The spie chart represents the global number of observed vs expected insertions inside and outside genes. The area of a wedge is proportional to the observed value, and its angle is proportional to the expected value (so depleted categories are within the gray circle and enriched categories protrude outside). The numbers represent observed over expected insertions, expressed in thousand. **d** Insertion sites in repeats. The bar plot shows the proportion of barcodes mapping to repeated sequences (see Table 1)

from intergenic regions with a 20% deficit over random. In conclusion, insertion biases toward transcribed chromatin are present but minor.

The inserted reporters also allowed us to interrogate repeated sequences (Fig. 2c), but not all the barcodes could be mapped, so most of the repair events occurred at unknown or ambiguous locations (Table 1). Yet, the subset that was mapped shows that the integrated reporters cover the mouse genome with sufficient uniformity to study regional repair biases.

### Mismatch repair on the reporters is strand-biased

To quantify the outcome of DNA repair on the integrated reporter, we designed two sequencing assays based on unique molecular identifiers (UMIs). The DNA extracted from the cell pools was first digested with I-SceI in order to eliminate the reporters that were not cleaved in ES cells—repairing the double-strand break through either NHEJ or SSA destroys the I-SceI restriction site. Using primers decorated with UMIs, we then performed either 50 cycles of linear amplification or 6 cycles of PCR (UMI-LA or UMI-PCR, Fig. 3a). The reasons for using UMIs are twofold: First, they make the quantification more robust (random fluctuations in the first PCR cycles can have large effects on the read numbers). Second and more importantly, they were used to mitigate template switching [27], a common PCR artifact that can potentially shuffle the barcodes between templates and make quantification inaccurate. The UMIs were inserted opposite to the barcode, so that we could discard UMIs associated with multiple barcodes (those can

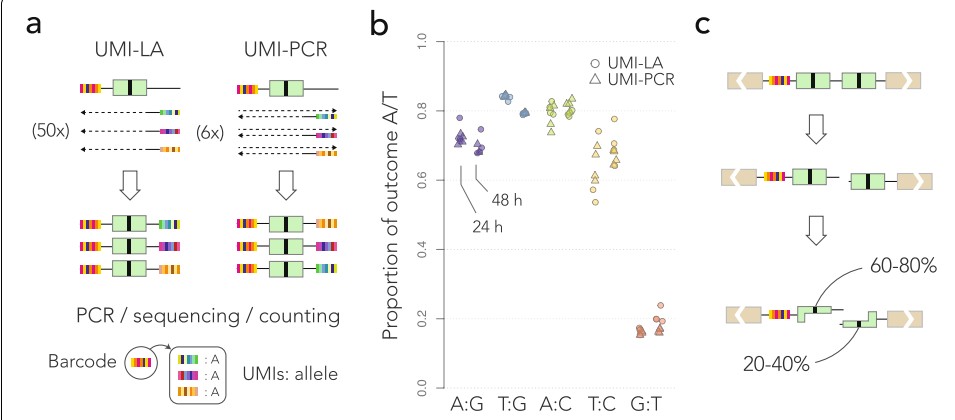

**Fig. 3** Measure of repair biases. **a** Quantification methods. In UMI-LA (left), barcoded reporters are amplified by 50 cycles of linear amplification using a primer decorated by UMIs. In UMI-PCR (right), reporters are amplified by 6 PCR cycles where one primer is decorated by UMIs. Either way, the products are further amplified by regular PCR. After sequencing, each barcode is associated with several UMIs, themselves associated with alleles. Repair biases are quantified by giving one vote per UMI. **b** Repair outcome. The dot plot shows the measured bias toward A or T (whichever applies) in each technical replicate. For each construct, data points obtained 24 and 48 h post I-SceI induction are shown on the left and on the right, respectively. **c** Graphical summary of the results. The nucleotide of the top strand is most frequently kept during the resolution of the mismatch

occur only through template switching). Counting UMIs thus provides more accurate measurements than counting sequencing reads.

For each construct, we performed 4 technical replicates per UMI-LA and per UMI-PCR, at 24 and 48 h post-I-SceI induction, plus 4 technical replicates with UMI-LA and 4 with UMI-PCR for each construct without I-SceI induction. Two UMI-PCRs for the A:G construct at 48 h failed to yield any sequencing read. Figure 3b shows the global repair bias toward A/T for each construct (the barcodes detected without I-SceI induction are discarded, see below). This represents a total of almost 330,000 repair events from mapped and unmapped barcodes (Table 1). Mismatch repair was reproducibly biased in all the tested conditions, with a bias in the range 60–80% toward A/T for the first four constructs, and around 20% for the last. For each construct, the biases were similar between replicates, between amplification methods and between time points, showing that the assays are reproducible in the given experimental conditions.

Strikingly, the dominant outcome did not correspond to a nucleotide but to a strand. Indeed, the T:G mismatch was resolved in favor of T in the T:G construct (cyan, Fig. 3b), but in favor of G in the G:T construct (red, Fig. 3b) where the nucleotides were swapped. The measured repair biases are roughly symmetric between the two constructs (80 vs 20%). We therefore conclude that in this assay, the top strand is more likely to be used as a template during the repair of the mismatch.

The magnitudes of the repair bias in favor of A or T when they are on the top strand (purple, cyan, green, and yellow, Fig. 3b) are comparable to each other, except for the T:C construct, which showed greater variations (we obtained fewer events for this construct, see Table 1). This means that in the present context, the nature of the mismatch has less influence than the nature of the allele on the top strand. Taken together, these results suggest that mismatch repair during SSA can be strongly biased toward a strand, regardless which nucleotides are mismatched (Fig. 3c).

## All the reporters have similar strand biases

Is the 60–80% bias a typical value for most reporters or a conflated average? We addressed this question in several steps by leveraging the properties of the UMI-amplicons.

First, we made sure that the reporters were cleaved by I-SceI. Here, we took advantage of the fact that the same barcode is sometimes found in different UMI-LA or UMI-PCR replicates. The repair events observed without I-SceI induction tend to be identical among replicates (Fig. 4a). Those represent barcodes from reporters where the F segments have recombined earlier than the I-SceI induction (e.g., during the preparation of the barcoded library), they are replicated through cell division and therefore are all identical. In contrast, the repair events observed 24 and 48 h post-I-SceI induction can differ between replicates, even if they take place on identical integrated reporters. This shows that the repair events observed upon I-SceI induction are distinct from the spurious repair events that took place earlier.

Second, we ruled out that the mismatches are unrepaired at the time they are assayed. If a mismatch is not repaired, the two strands are not complementary so UMI-LA and UMI-PCR give different results (Fig. 4b). UMI-LA uses only one strand as a template (the top one), so all the associated UMIs must report the same outcome for a given barcode, even when the mismatch is not repaired. In contrast, UMI-PCR uses both strands as a template, so the UMIs are associated with both nucleotides of the unrepaired mismatch. Therefore, the results of UMI-LA and UMI-PCR should be strongly discordant for unrepaired mismatches.

As noted in Fig. 3b, UMI-LA and UMI-PCR are concordant, suggesting that the mismatches are repaired. To confirm this conclusion, we measured the proportion of barcodes with conflicting UMIs, i.e., UMIs reporting different alleles (Fig. 4c). UMI-PCR did

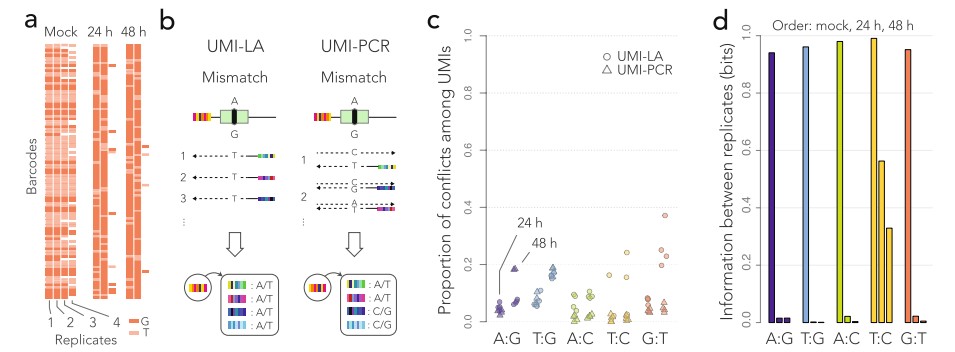

**Fig. 4** Global differences between reporters. **a** Repair across replicates. Each row shows a random barcode from the G:T construct, and each column shows a replicate where it appears. The color of the rectangle indicates the repair outcome. Without I-SceI induction (Mock), barcodes are typically present in more than two replicates, always with the same outcome. Twenty-four or 48 h post-I-SceI induction, barcodes are typically present in two replicates with different outcomes. The results are similar for all the constructs (not shown). **b** Amplification of unrepaired mismatches. If the mismatch is unrepaired, UMI-LA (left) produces UMIs with the allele of the top strand only, whereas UMI-PCR produces UMIs with both alleles. **c** Conflicts among UMIs. The dot plot shows the proportion of barcodes such that at least one UMI reports a different allele than the majority. Colors and symbols are the same as in Fig. 3b. **d** Mutual information between replicates. The bar graph shows the average mutual information per barcode between pairs of replicates (experiments with the T:C construct had only 100 pairs, compared to > 1000 pairs for the other constructs and the no I-SceI controls)

not produce more conflicts than UMI-LA, confirming that the repair biases observed in Fig. 3b apply to fully repaired mismatches.

Note that in Fig. 4c, the great majority of barcodes do not have a single conflicting UMI. This suggests that the input DNA in UMI-based assays consists of just one molecule per barcode. The alternative would be that copies of the same reporter are consistently repaired the same way, i.e., that reporters have distinct predefined biases and that the 60–80% bias is a conflated average.

To distinguish between these two hypotheses, we computed the mutual information between repair outcomes on the same barcodes (see the "Methods" section). For categorical variables, mutual information is more adequate than the Pearson and Spearman coefficients, and the interpretation is similar in the sense that a value of 0 indicates that the variables are independent. We can thus use this metric to test whether reporters have individual biases: if they do, the repair outcome is partly determined by this individual bias, so knowing how a reporter is repaired in one replicate gives some information about how it is repaired in other replicates, yielding a nonzero mutual information.

We collected the barcodes appearing in at least two replicates and assigned them to a single dominant repair outcome. For each construct and each time point, we filled a 2x2 contingency table with all the replicate pairs, from which we computed the mutual information shown in Fig. 4d. Without I-SceI induction, the mutual information between replicates is close to 1, meaning that the reported outcome is always the same for a given barcode, consistently with Fig. 4a (recall that those repaired reporters were replicated through cell division). In contrast, the mutual information drops to 0 for barcodes that are amplified after I-SceI induction. This means that the repair of a reporter in one replicate has no predictive value for the repair in another replicate. In other words, if a reporter is repaired toward A, say, it is not more likely to be repaired toward A in other experiments (as apparent in Fig. 4d). Therefore, there exists no group of reporters with a much higher (or lower) bias than average.

Taken together, these results show that in the conditions of our assay, the mismatches occurring through SSA are repaired with a ubiquitous bias toward the top strand. In other words, regardless of their location, the reporters all tend to repair the mismatch toward this outcome.

### The insertion site has a weak influence on the repair bias

Our results indicate that mismatch resolution on the reporters has a global bias toward the top strand, but does this bias change with the genomic context? Fig. 5a shows the repair bias of the G:T construct (that with the highest number of mapped events) when the reporters are inserted in expressed genes, silent genes, or intergenic regions. In all the cases, the bias is toward G and it has the same magnitude as the genome-wide average (Fig. 3b). Figure 5b shows a similar analysis for the histone marks at the insertion site. The bias toward G is again comparable to the genome-wide average, with a variance that depends on the number of reporters in each category (from 0.8% coverage for H2Aub1 and H4K20me3, to 5% coverage for H3K36me3). Another factor influencing the variance is batch effect, which can be removed in order to lower the noise. A two-way ANOVA reveals that reporters inserted in the vicinity of H3K36me3 or H3K79me2 have a less pronounced bias (two-tailed Student's *t* test for coefficients in linear models, 164

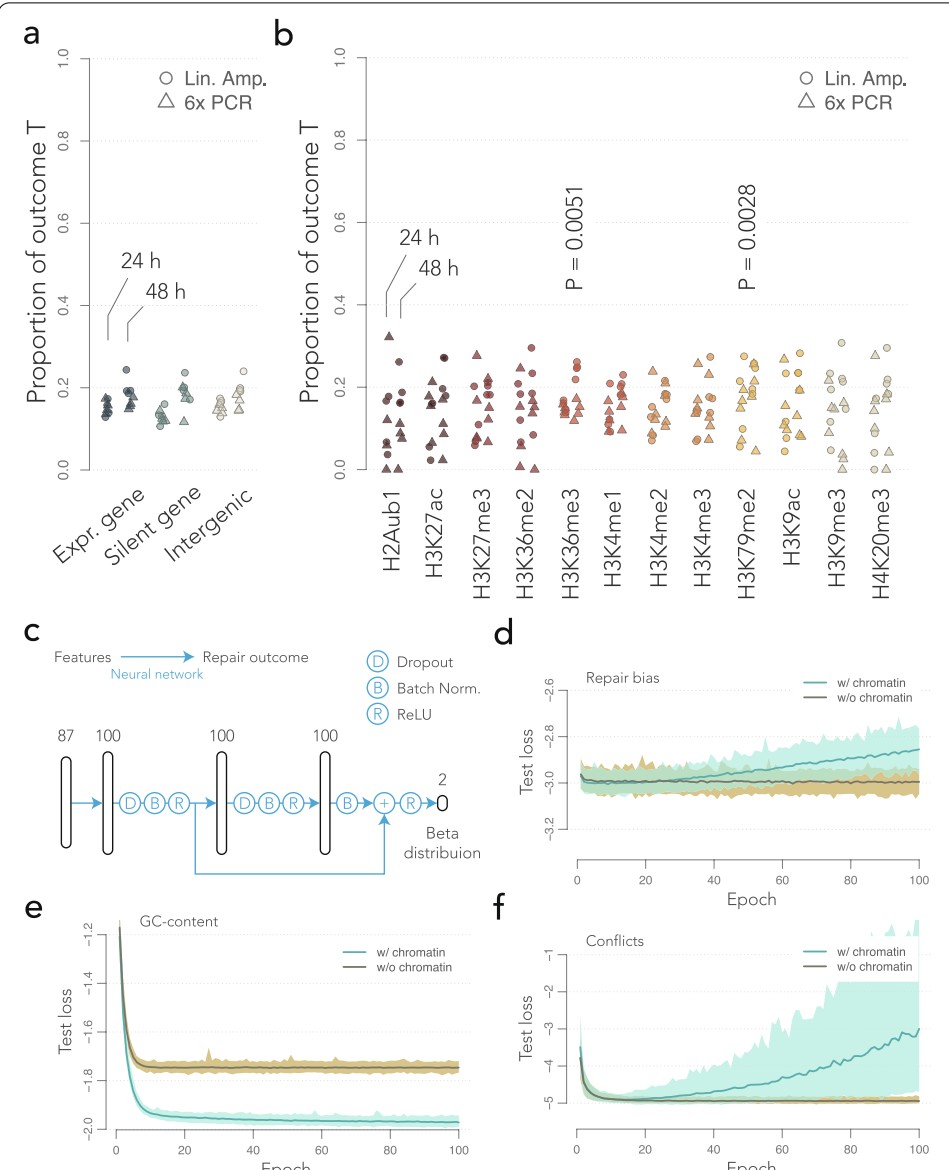

**Fig. 5** Mismatch resolution and chromatin context. **a** Biases in genomic contexts. The dot plot shows the measured bias toward T with the G:T construct. Expressed genes, silent genes, and intergenic regions are the same as in Fig. 2c. **b** Biases in chromatin contexts. Same representation as in **a** for chromatin marks. The P-values of a two-tailed Student's *t* test for coefficients in linear models are indicated if they are below 0.01. **c** Architecture of the artificial neural network. Vertical cartridges represent neuron layers with indicated dimensions. The blue arrows indicate the forward information flow. **d** Learning curves for the full data set. The median loss on the test set is shown in green for 100 trainings with random restart. The interval between the 1st and 99th percentile is shown by the green shaded area. The median loss for the null model without chromatin features is shown in brown, and percentiles are shown by the brown shaded area. **e** Learning curves for the local GC content. The repair outcome was replaced by the GC content in a 10 kb window around the reporter. **f** Learning curves for conflicts among UMIs. The only observations used for learning are UMI-PCR 24 h post-I-SceI induction

degrees of freedom, P = 0.00509, and P = 0.00275, respectively). The deviation is 2–3% toward T in both cases, i.e., substantially smaller than the baseline bias toward G. Interestingly, both histone marks have a direct role in DNA repair, though in different pathways [28]. H3K36me3 is involved in the mismatch repair pathway (MMR) by recruiting

Msh2–Msh6, while H3K79me2 is involved in the 53BP1 pathway to repair double-strand breaks and associated mismatches. Removing batch effects on the data of Fig. 5a does not reveal any statistically significant difference.

Such analyses focused on one histone mark at a time are bound to underestimate the real contribution of chromatin, which may involve combinations of marks. We thus used a flexible machine learning approach to evaluate how well the information about chromatin predicts the local repair bias. We designed an artificial neural network with a residual network architecture [29]. In a nutshell, the output of hidden layers is batch-normalized [30] and passed through a standard ReLU activation function [31]. We also included two Dropout steps to mitigate overfitting [32]. Finally, the output layer projects onto the parameters of a Beta distribution, so that the network can be used to predict proportions between 0 and 1 [33]. The architecture of the network is sketched in Fig. 5a (see the "Methods" section for details).

The dataset consists of 43,864 mapped repair events for which the chromatin features at the insertion site are available. Each record consists of the repair outcome for one barcode together with the local GC content and the chromatin features compiled by Juan et al. [34]. If a barcode was present in several replicates, the outcomes were considered to be independent events within the same chromatin context. The chromatin features include 3 cytosine modifications, 13 histone marks, and 62 chromatin proteins. Chromatin features are mapped at a resolution of 200 bp, and each record was constructed by inheriting the full chromatin profile of the window where the reporter is inserted. The records also include the construct type, the time point, the amplification technology, and the TRIP pool. Those bookkeeping variables were introduced to buffer non-biological variations such as batch effects. A random set of 10% of the records was held out for testing (see "Methods" section).

We performed 100 independent trainings with random restarts and tracked the performance with a loss function measuring the discrepancy between predicted and observed outcomes of repair. The learning curves represent the mean value of the loss function on the test set as learning unfolds: As long as the score goes down, the capacity of the model to predict the outcome on new data improves; when the loss stabilizes, the model reaches its maximum performance; and if the loss increases, the model is overfitted and further training damages the performance.

The median performance of the network on the test set is shown in Fig. 5d. For comparison, we included 100 trainings of a null model where the chromatin features were removed from the predictors, leaving only the construct, the time point, the amplification technology, and the TRIP pool. In other words, the null model cannot fit any context-dependence for repair events; it can only learn a mean and a variance per replicate, producing predictions of minimum quality. Surprisingly, the learning curve of the full model is above that of the null model, indicating that it is not more accurate than this minimum. In addition, the full model shows clear signs of overfitting from 30 epochs. It is important to highlight that the model was unable to pick up the effects of H3K36me3 and H3K79me2, probably because they are too small given the amount of available data (variations of 2–3% in ~5% of the cases).

To make sure that this strategy can reveal the influence of the context, we used the neural network for the comparable task of predicting the local GC content within a

10-kb window. In this case, the full model clearly outperforms the null model and still shows some evidence of learning after 100 epochs (Fig. 5e), demonstrating that the model can discover complex associations between the local chromatin context and other variables (the score of the null model improves in the first 5 epochs because it learns that the variations of the GC content do not span the full dynamic range between 0 and 1). In combination with the previous results, this suggests that the model can detect complex relationships, but not small effects.

Finally, we asked whether the occurrence of repair itself depends on the chromatin context. To address this question, we took advantage of a fact established previously: UMI-PCR is expected to produce mixed populations of UMIs if the mismatch is not repaired (Fig. 4b). We used the neural network of Fig. 5b to predict whether a barcode will have conflicting UMIs, when assayed by UMI-PCR 24 h post-I-SceI induction (total of 8734 records, Fig. 5f). As for the repair bias, the learning curve of the null model is below that of the full model. This means that the neural network could not find a potent influence of the chromatin context on the probability of repairing the mismatch.

The unmapped reporters were excluded from this analysis, so it is possible that repair proceeds differently in some unmapped regions, such as the Y chromosome for instance. Otherwise, our results suggest that the insertion site of the reporter has a subtle influence on the outcome of mismatch repair. We could detect local variations of 2–3%, but we could not reliably train a deep-learning model to predict the repair outcome.

### An assay to infer the methylation status of the reporters

We have assumed that upon integration, the reporter does not modify the local chromatin context. However, there are cases when this does not hold. For instance, vectors based on murine leukemia viruses (MLVs) are silenced by dedicated transcription factors that set a repressive chromatin environment on the construct [35]. Our TRIP reporter was designed to avoid similar scenarios: First, the Sleeping Beauty transposon evolved in fish [36], minimizing the chances for co-evolutionary adaptations in mouse; second, the Sleeping Beauty backbone is transparent to position effects [37]; and third, the mismatch was embedded in the sequence of the widely used reporter gene GFP.

However, these precautions are no proof that the backbone is truly neutral vis a vis the chromatin of the insertion site. To test this, we took advantage of a casual observation: the constant parts of the F segments often had CG>TG transitions, reminiscent of the mutation pattern of methylated CpGs [8]. DNA methylation is an important indicator of the local epigenomic state, so we set out to test whether the sequence flanking the mismatch could be used to infer the methylation status of the integrated reporters.

To figure this out, we designed a control experiment using custom oligonucleotides with a methylated or non-methylated CpG, flanked by random nucleotides forming a split barcode (Fig. 6a). We amplified those oligonucleotides by UMI-LA using the same primer and the same conditions as the TRIP reporters. If at some point during the 50 cycles of the reaction, the methylated (or non-methylated) cytosine is spontaneously deaminated, it turns into a thymine (or a uracil) that pairs with adenine in all subsequent cycles, creating a discordant pool of UMIs that can be used as a "smoking gun" for deamination (Fig. 6a, see the "Methods" section).

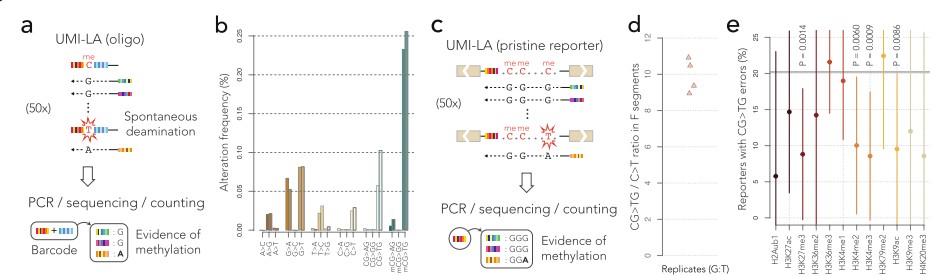

**Fig. 6** Using CG>TG transitions during UMI-LA to infer methylation status. **a** Calibrating the deamination assay with a positive control. UMI-LA is performed on barcoded methylated oligonucleotides. Methylated cytosine can spontaneously deaminate at high temperature and be converted into thymine, thus producing conflicting UMIs. **b** Frequency of alterations during UMI-LA. The bar graph shows the percentage of barcodes with associated alterations found among their UMIs. **c** Principle of the assay. Uncut reporters in control experiments contain 19 CpGs that can be assayed as in **a**. **d** Dot plot showing the ratio of CG>TG transitions inside vs outside the CpGs of the F segments for four replicates. The average around 10 matches the 10-fold increase observed for methylated CpG in panel **b**. **e** Proportion of pristine reporters with CG>TG transitions in different chromatin contexts. Histone marks present at the insertion site are indicated on the *x*-axis. Dots indicate the local mean and vertical bars show the limits of a 99% confidence interval for this mean. The horizontal bar shows the global average of reporters with CG>TG transitions (the value around 20% is consistent with the baseline of Fig. 6b, taking into accounts that there are 19 CpG and 4 pooled replicates). *P*-values below 0.01 are indicated on the graph (two-tailed Student's *t* test to compare proportions, *n* = 2182). Significant reductions of CG>TG transitions are observed in H3K27me3, H3K4me2, H3K4me3, and H3K9ac, all associated with active promoters in mouse ES cells

Figure 6b shows the proportion of barcodes with discordant UMIs for different types of alterations occurring during the linear amplification. The C>T transition has remarkably distinct frequencies in different contexts: it is 2–4 times more frequent in non-methylated CpGs than outside CpGs, while in methylated CpGs, the fold increase rises to approximately 10. Those values are compatible with previous estimates of the rate of deamination at high temperature in vitro [38]. Note that the estimates are substantially lower than the error rate of oligonucleotide synthesis: this is possible because synthesis errors do not produce discordant UMI pools and are therefore excluded, demonstrating yet another advantage of using UMIs.

The results above suggest that it is possible to infer the methylation status of CpGs by comparing the frequency of C>T errors inside vs outside CpGs. The ratio should be 2–4 for non-methylated CpGs and around 10 for methylated CpGs. We thus focused on the G:T construct of Fig. 5, using control experiments where the reporters are neither cut nor repaired in order to rule out possible interferences from the repair process (Fig. 6c). The C>T transitions were approximately 10 times more frequent inside CpGs than outside (Fig. 6d), suggesting that most CpGs in the reporters of those experiments are methylated.

In mammalian genomes, CpGs are normally methylated, except in rare regions that usually coincide with regulatory sequences [8]. To test whether reporters also follow this pattern, we measured the proportion of reporters with CG>TG transitions in different regions. This metric has a low signal-to-noise ratio because CG>TG transitions in methylated CpGs are only three times more frequent than in non-methylated CpGs (Fig. 6b). However, the high number of inserted reporters gives substantial statistical power to detect local trends. Figure 6e shows the proportion of reporters with CG>TG transitions in different chromatin contexts. At the significance level of 0.01, four histone marks are

associated with a reduction of CG>TG transitions. The marks H3K4me2, H3K4me3, and H3K9ac are considerably enriched on promoters and mutually exclusive with CpG methylation [39, 40]. Interestingly, the mark H3K27me3 is a feature of heterochromatin in differentiated cells, but in ES cells, it is found on the so-called bivalent promoters [41]. Reinforcing this trend, reporters inserted in the promoters of active genes show fewer CG>TG transitions (data not shown, two-tailed Student's $t$ test to compare proportions P = 5.2 $10^{-6}$, 99% confidence interval for the difference 8–26%, $n = 2182$). Taken together, these results suggest that reporters inserted in promoters are protected from DNA methylation.

Overall, this analysis shows that UMI-LA captures a signature for CpG methylation. Furthermore, it suggests that the methylation of integrated reporters is context-dependent and that it reflects existing variations in the genome of mouse ES cells. This is evidence that the chromatin where repair takes place is not determined by the sequence of the reporter, but by the context at the insertion site.

### Asymmetries between flaps influence the repair bias

The results of our repair assay show that one strand is favored during mismatch resolution. What could create such an asymmetry between the strands? The position of the I-SceI site is skewed toward one of the F segments, meaning that after the resection of 5′ ends, one 3′ flap is longer than the other. Could this cause the repair bias observed throughout the genome?

To test this hypothesis, we used the CRISPR-Cas9 system to change the position of the double-strand break. I-SceI leaves a 3′ flap of 33 nucleotides on the top strand, and 13 nucleotides on the bottom strand. We designed a control guide RNA that induces a double-strand break near the I-SceI site, leaving 3′ flaps of 31 and 11 nucleotides, respectively. We also designed a second guide RNA that induces a double-strand break at a symmetrically opposite location, leaving 3′ flaps of 0 and 42 nucleotides respectively (Fig. 7a). If the asymmetry between the flaps induces the repair bias, one expects that the first guide RNA should induce the same bias as I-SceI, whereas the second guide RNA should induce a bias in the opposite direction.

Figure 7b suggests that this is indeed the case. The G:T mismatch is repaired in favor of G when initiating the double-strand break with I-SceI or guide RNA #1 and in favor of T when initiating it with guide RNA #2. To test whether the bias can be reverted on the exact same reporter, we collected all the mapped barcodes for which the repair was measured for both guide RNAs. We then used a color code to depict both outcomes simultaneously (Fig. 7c). The results show that the bias is reversed on individual reporters when replacing guide RNA #1 with guide RNA #2. The switch is uniform throughout the genome, consistent with the previous finding that the insertion site of the reporter has little effect on the resolution of the mismatch.

Taken together, our results show that the position of the double-strand break can bias the mismatch resolution during SSA. More specifically, in our repair assay, the asymmetry between the 3′ flaps results in a repair bias where the strand with the longest flap is more likely to be used as a template.

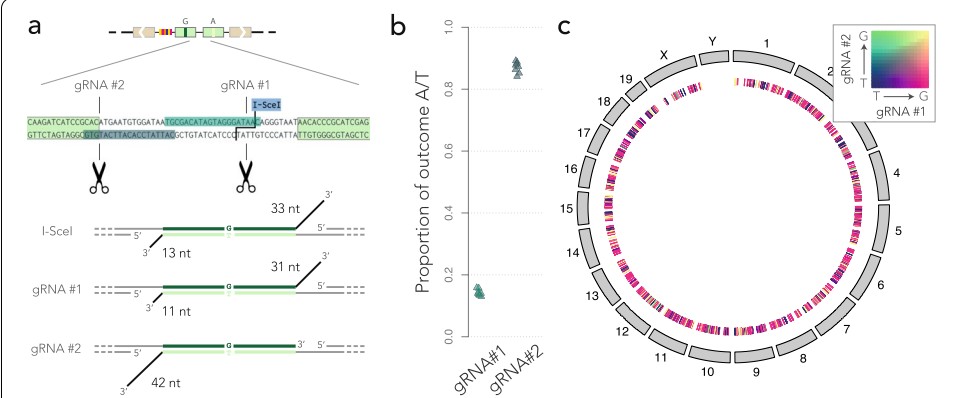

**Fig. 7** Mismatch resolution and flap asymmetry. **a** Sketch of the construct and of the guide RNAs used to induce double-strand breaks at alternative locations. **b** Global bias reversal. The dot plot shows the bias toward T with the G:T construct, measured by UMI-PCR 24 h after inducing a double-strand break with the guide RNAs shown in **a**. **c** Local bias reversal. The circular map of the mouse genomes shows a random sample of 1000 inserted G:T reporters where the final state after repair was known with both guide RNAs shown in **a**. Each tick mark represents an integrated reporter, and its color represents the outcome of both experiments, as per the heat map in the top right corner. Most tick marks are magenta, indicating that the reporter was repaired toward G when using guide RNA #1 and toward T when using guide RNA #2

## Discussion

Here, we used a TRIP assay to study the process of DNA repair in the chromatin of mouse ES cells. Our construct is designed to produce a mismatch if the reporter is repaired through the single-strand annealing (SSA) pathway, allowing us to study how the same mismatch is repaired at different loci. With TRIP reporters inserted throughout the genome (Fig. 1 and Table 1), we obtained a global landscape of the biases of mismatch resolution. We found no evidence that repair is intrinsically biased toward G and C nucleotides. Instead, we found a persistent 60–80% bias toward the strand with the longest 3′ flap (Fig. 6), regardless of the mismatch that was induced. We also observed that the repair bias is uniform throughout insertion sites, suggesting that the chromatin surrounding the lesion has little influence in this context. Overall, these results have important implications regarding the factors influencing the repair of mismatches.

It is challenging to tease apart the individual contributions of DNA damage and repair to mutational processes. This has been possible on plasmids [15], but the known interactions of mismatch repair with chromatin suggest that mutational biases should be studied directly in the genome [42]. In this regard, the TRIP assay developed here is a technical step forward. A similar principle was already used by Gisler et al. [24] and Schep et al. [43]. Interestingly, the authors found that the local state of chromatin influences the choice of repair pathways, which we did not investigate in this study.

We found that the chromatin at the insertion site had a small effect on the repair bias. The G:T reporters inserted in the vicinity of the histone marks H3K36me3 and H3K79me2 had a bias toward G that was 2–3% smaller (Fig. 5b). Both marks are directly involved in DNA repair [28], and they overlap to a large extent. Either the mismatches are repaired less efficiently, or those histone marks alter the bias. The lack of repair can be evaluated by the discordance between UMI-LA and UMI-PCR. While such a difference is apparent for H3K36me3, it is also the case at the genome-wide scale (Fig. 3b) and

we have established that lack of repair is probably not the reason (Fig. 4). Interestingly, this construct produces a CG:GT mismatch, so if the cytosine is methylated, the mismatch is a substrate for Mbd4, which catalyzes the removal of the thymine. Therefore, it is possible that H3K36me3 and H3K79me2 interfere with the Mbd4 pathway. Further work will be needed to fully understand why the bias is slightly different in those regions.

Using a particularity of the UMI-LA, we brought evidence that the methylation of integrated reporters is not uniform in the genome (Fig. 6). This suggests that the insertion site has an influence on the chromatin context in which DNA repair takes place. Yet, the bias is relatively stable throughout the genome. This conclusion is in line with the fact that the measured biases are substantially larger than the typical effects of chromatin on mismatch repair [44]. Due to the inherent limitations of using reporters, the full contribution of chromatin will best be established in situ; for instance by inducing double-strand breaks between naturally occurring tandem repeats (e.g., using the strategy shown in Fig. 7).

The mismatches produced in our repair assay are coupled to the repair of a double-strand break. In that sense, they differ from the typical mismatches that occur during DNA replication. We nevertheless gained general insight into the mechanism of mismatch repair, the most striking of which being that the process can be substantially strand-biased. Here, it is worth mentioning that the asymmetry between the strands was not intended in our experimental design; the serendipitous discovery that it was due to the skewed location of the double-strand break was as described in the results. In hindsight, the bias reversal is a strong internal control that the mismatches are repaired within 24 h and that our experimental system can detect a skew in either direction.

It was recently discovered in yeast that the flaps influence the efficiency of mismatch repair in break-induced replication [45], so there exists a cross-talk between flap excision and mismatch repair. In addition, repair in a common yeast assay to study SSA also shows an approximately 70% bias toward the strand with the longer flap [46]. Importantly, the work showing that there is no repair bias except on G:T mismatches in rodent cells was based on the formation of heteroduplexes without flaps [16]. So overall, our results are in line with previous experiments in the field.

We have not established experimentally that the flaps are exactly as shown in Fig. 6a, but the repair mechanism of SSA is well enough established that we can accept the model as a starting point for further discussion. In particular, it is interesting to speculate as to how the flaps may influence the choice of the nucleotide to be excised. The repair of the mismatch must be concomitant with that of the double-strand break, and the fact that one strand is favored suggests that there is a race condition between the processing of the flaps. In the experiment with guide RNA #2, the disfavored strand has no flap at all, which suggests that the asymmetry is in the timing of flap removal. We thus propose the "first-nick" model depicted in Fig. 8. The flaps prevent strand discrimination by the MSH complex; when a flap is removed, a nick or a breach is exposed on the associated strand, providing the signal needed by MSH2-MSH6 to backtrack and resolve the mismatch [47]. At that stage, the involvement of the mismatch repair system has not been established experimentally and the central postulate of our model is that it takes longer to remove long flaps than short flaps. Whether this is the case will require further investigations.

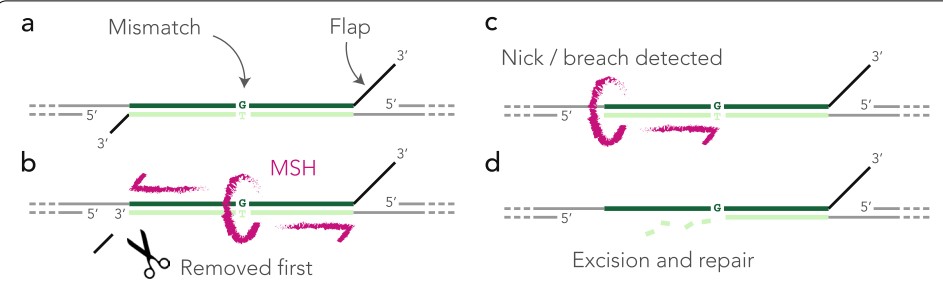

**Fig. 8** First-nick model. **a** After resection of the 5′ ends and annealing of the complementary strands, unannealed flaps extend in 3′. **b** The MSH complex recognizes the mismatches and slides to initiate strand discrimination but the flaps jam the process. Shorter flaps tend to be removed earlier, allowing for strand discrimination. **c** The nick or breach exposed by the removal of the flap is recognized by the MSH complex that backtracks to initiate the repair of the mismatch. **d** The mismatch is repaired as usual by excising and re-synthesizing the strand

Using the properties of UMI-PCR, we could establish that the mismatches were repaired at the time of the assay. The reporter cannot be amplified by PCR before the flaps are removed and the gaps are sealed (the primers would not anneal to the template). This suggests that the mismatch is repaired before the double-strand break is fully sealed or shortly thereafter, in line with the first-nick model. In any event, the location of the reporter did not seem to influence whether the mismatch is repaired, in apparent contradiction with the well-established fact that mismatches in late-replicating regions are repaired less efficiently [20, 21, 48]. It may be that mismatches occurring through SSA are more accessible because of the double-strand break. Alternatively, the unique chromatin features of ES cells may favor widespread mismatch repair throughout the genome [49]. But this study raises the possibility that a different mechanism is at work: mismatches in late-replicating regions may be repaired with the same probability, but with less accurate strand discrimination. Our model suggests that unannealed regions such as those arising at the convergence of replication forks [50] can mask downstream nicks and obfuscate strand discrimination.

## Conclusions

The work presented here identifies the molecular features of a newly identified strand-bias in DNA repair. We found no evidence that any nucleotide was intrinsically favored in our assay. Instead, we found that the molecular details of the damage such as the position of the cut and the length of the 3′ flaps had a large influence on the outcome of mismatch resolution.

## Methods

### Plasmid construction and library preparation

Plasmid pCBASceI for I-SceI expression and plasmid pcDNA3.1-mCherry were obtained from Addgene (#26477 and #128744 respectively). Guide RNAs TGCGACATA GTAGGGATAAC (gRNA1) and GCATTATCCACATTCATGTG (gRNA2) were cloned into plasmid pSpCas9(BB)-2A-GFP (Addgene #48138) by the company VectorBuilder and shipped as pRP[CRISPR]-EGFP-hCas9-U6> {VP_gRNA1} and pRP[CRISPR]-EGFP-hCas9-U6> {VP_gRNA2}, with internal identifiers VB200402-2406dex and

VB200402-3189qpa, respectively. Plasmid pCMV(CAT)T7-SB100X for Sleeping Beauty 100X expression was kindly provided by Zsuzsanna Izsvák, plasmid pcDNA3.1-mCherry. FF fragments (each with a precursor for one of heteromismatches) were synthesized by Life Technologies and cloned into plasmid pT2 using Gibson Assembly Cloning Kit (NEB, E5510S). Obtained pT2_FF plasmids were used as templates for PCR-based barcoded library preparation [23].

For barcoding PCR, 100 pg of each of pT2_FF plasmid was used as template in 50 μL Phusion DNA polymerase reaction mix (Thermo Fisher Scientific, F530S) with GC buffer, using PCR primers L1-6 from Table 2 in the following cycling conditions: 98 °C for 1 min; 98 °C for 30s, 60 °C for 30s, and 72 °C for 3 min (25 cycles); and 72 °C for 5 min. The template was destroyed by adding 1 μL 20,000 U/mL DpnI (NEB, R0176S) to the mix and incubating at 37 °C for 1 h. The products were purified with a QIAquick Gel Extraction Kit (Qiagen). For T:G-library preparation, the PCR product was self-ligated with T4 DNA ligase (Thermo Fisher Scientific, EL0013) with 5% PEG 4000 at 4 °C overnight. For other three libraries, PCR products were digested using NheI

**Table 2** List of primers

| Primer name | Sequence |
| --- | --- |
| L1 | NNNNNNNNNNNNNNNNNNNNagatcggaagagcgtcgtg |
| L2 | CGCTAATTAATGGAATCATGAACA |
| L3 | catagGCTAGC NNNNNNNNNNNNNNNNNNNNagatcggaagagcgtcgtg |
| L4 | catagGCTAGC TCCGCAGAATCATGAACA |
| L5 | catagGCTAGC AGTCAGGAATCATGaaca |
| L6 | catagGCTAGC TCGTTGGAATCATGaaca |
| IR1 | TGTATTTGGCTAAGGTGTATGTAA |
| IR2 | ATTCCCAGTGGGTCAGAAGT |
| M1 | AATGATACGGCGACCACCGAGATCT ACACTCTTTCCCTACACGACGCTCTTCCGATCT |
| M2 | CAAGCAGAAGACGGCATACGAGATCGGTCTCGGCATTCCTGCTGAACCGCTCTTCCGATCT ACT AAA CTTCCGACTTCAACTGT |
| M3 | CAAGCAGAAGACGGCATACGAGATCGGTCTCGGCATTCCTGCTGAACCGCTCTTCCGATCT TGT AAA CTTCCGACTTCAACTGT |
| U1 | GTGACTGGAGTTCAGACGTGTGCTCTTCCGATCTNNNNNNNNNNNNNNNNNNNNTGCAACGAATTCATTAGTGCG |
| U2 | AATGATACGGCGACCACCGAGATCTACACTCTTTCCCTACACGACG |
| UIND1 | CAAGCAGAAGACGGCATACGAGAT CAAGCT GTGACTGGAGTTC |
| UIND2 | CAAGCAGAAGACGGCATACGAGAT GTGAAC GTGACTGGAGTTC |
| UIND3 | CAAGCAGAAGACGGCATACGAGAT ACTGCA GTGACTGGAGTTC |
| UIND4 | CAAGCAGAAGACGGCATACGAGAT CTAAGA GTGACTGGAGTTC |
| UIND5 | CAAGCAGAAGACGGCATACGAGAT CTTGGC GTGACTGGAGTTC |
| UIND6 | CAAGCAGAAGACGGCATACGAGAT AGACAT GTGACTGGAGTTC |
| UIND7 | CAAGCAGAAGACGGCATACGAGAT TGTAGA GTGACTGGAGTTC |
| UIND8 | CAAGCAGAAGACGGCATACGAGAT TTCAGC GTGACTGGAGTTC |
| UIND9 | CAAGCAGAAGACGGCATACGAGAT GTCCTA GTGACTGGAGTTC |
| UIND10 | CAAGCAGAAGACGGCATACGAGAT ATCCAG GTGACTGGAGTTC |
| UIND11 | CAAGCAGAAGACGGCATACGAGAT ACATCG GTGACTGGAGTTC |
| UIND12 | CAAGCAGAAGACGGCATACGAGAT GCCTAA GTGACTGGAGTTC |
| METH-Y | ACACTCTTTCCCTACACGACGCTCTTCCGATCTBDHVBDHVBDHVGATMGATCBDHVBDHVBDHVCGCACTAATGAATTCGTTGC |
| METH-N | ACACTCTTTCCCTACACGACGCTCTTCCGATCTBDHVBDHVBDHVGATCGATCBDHVBDHVBDHVCGCACTAATGAATTCGTTGC |

restriction enzyme (NEB, R0131S) at 37 °C for 3 h and self-ligated with T4 DNA ligase (Thermo Fisher Scientific, EL0013) with 5% PEG 4000 at 4 °C overnight. Ligated products (100–400 ng/μL) were desalted by drop dialysis using 13-mm diameter, type-VS Millipore membrane (Merck Millipore, #VSWP01300). Twenty microliters of Electro-MAX DH10B competent cells (Invitrogen, 18290015) was electroporated with 3-μL ligated products. 0.01% of the electroporated bacteria were plated on an ampicillin-containing medium in order to estimate the complexity of the libraries; the remaining cultures were grown overnight in 100ml of liquid medium, and the plasmids were extracted the next day. Barcoded plasmid libraries with a complexity of 0.8–2 million independent clones were used for further experiments.

### Cell culture

mESCs were grown at 37 °C under a 95% air and 5% $CO_2$ atmosphere on gelatin in serum/LIF medium composed of GMEM (Sigma, G5154) supplemented with 15% FBS (HyClone™), 1X MEM Non-Essential Amino Acids (Gibco, 11,140–050), 1X GlutaMax (Gibco, 35,050–061), 1mM sodium pyruvate (Gibco, 11360-070), 0.1 mM 2-Mercaptoethanol (Thermo Fisher, 31,350–010), and 1000 U/ml LIF ESGRO® (Merck Millipore, ESG1106). mESCs were passaged every 2 days with a 1:8 dilution. Cells were tested yearly for mycoplasma contamination.

### Transfection and transposon integration

To integrate the construct into the genome of mouse ES cells, 1 million cells in 6-well plates were transfected with 2 μg of plasmid pT2_FF together with 2 μg of plasmid pCMV(CAT)T7-SB100x and 2 μg of plasmid pcDNA3.1-mCherry using Lipofectamine 2000 (Thermo Fisher, #11668027). After 24 h, mCherry-positive cells were FACS sorted. Pools of 20,000 cells were plated on 24-well plates and grown for 2 weeks, transferring to 100-mm dishes when the cultures reached a density of $5x10^6$ cells/mL. Two independent cell pools of 20,000 cells were prepared for each construct.

To generate the mismatches, pools of mouse ES cells harboring integrated transposons were transfected with 5 μg of plasmid pCBASceI using Lipofectamine 2000 (Thermo Fisher, #11668027). The growth medium was changed after 16 h later. Twenty-four hours and 48 h post-transfection, cells were collected using Trypsin-EDTA (Gibco, #25200056), washed with PBS, and used for genomic DNA isolation. Genomic DNA from transfected cells were extracted using DNeasy Blood and Tissue Kit (Qiagen, #69504).

Mismatch generation via CRISPR-Cas9 was done similarly. Five micrograms of either plasmid pRP[CRISPR]-EGFP-hCas9-U6> {VP_gRNA1} or plasmid pRP[CRISPR]-EGFP-hCas9-U6> {VP_gRNA2} were used for the transfection. Transfected cells were collected 24h post-transfection.

### Inverse PCR

Ten micrograms of genomic DNA from transfected mESCs were digested using 10 μl 10 U/μL NlaIII (NEB, #R0125S) in a 50 μL final volume for 3 h at 37 °C. The reaction

was heat-inactivated at 65 °C for 20 min. The reaction was diluted to a final volume of 1.8 mL in 1X T4 ligase buffer (Thermo Fisher Scientific, #EL0013) to favor self-ligation events, and ligation was carried out with 600 U of T4 ligase (Thermo Fisher Scientific, #EL0013) at 16 °C overnight. After ligation, samples were precipitated by ethanol, pellets were resuspended in water and column-purified (QIAGEN, QIAquick PCR purification kit #28104) eluting with 100 uL EB. Non-circularized templates were eliminated by 2 h digestion at 37 °C with Plasmid-safe DNAse (Epicentre, #E3101K), and the enzyme was inactivated by heating for 30 min at 70°C. The product was column-purified (QIA-GEN, QIAquick PCR purification kit #28104). The backbone of the TRIP reporters contains a I-CeuI site outside the transposable cassette, taking advantage of this, non-integrated plasmids were cut by 2 h digestion at 37 °C with I-CeuI restriction enzyme (NEB, R0699S) in a total volume of 70 ul followed by 20 min heat inactivation at 65 °C. All enzymatic reactions were carried out in the recommended manufacturer's buffer.

For the first round of nested PCR, 10μL of I-CeuI-digestion products was mixed in 50 μL standard Phusion polymerase reaction mix (Thermo Fisher Scientific, F530S) in GC buffer, with 0.1 μM primers IR1,2 (annealing to IR/DR sequence). The cycling conditions were as follows: 98 °C for 5 min; 98°C for 20 s, 60°C for 1 min, and 72 °C for 5 min (1 cycle); 98 °C for 20 s, 60 °C for 1 min, and 72 °C for 2 min (20 cycles); and 72 °C for 5 min. Products of the reaction were purified using Agencourt AMPure XP beads (Beckman Coulter, A63880) and eluted in 40 μL of water. For the second round of nested PCR, 37 μL of the products was diluted to 50 μL of standard Phusion polymerase reaction mix in GC buffer with 0.1 μM primer M1 (annealing to the Illumina PE1.0 primer) and one of indexing primers M2,3. The cycling conditions were as follows: 98 °C for 2 min; 98 °C for 20 s, 60 °C for 1 min, and 72 °C for 1 min (10 cycles); and 72 °C for 5 min. Primers M2,3 added the Illumina PE2.0 primer and one of indices to the amplicons. Products of the reaction were purified using Agencourt AMPure XP beads (Beckman Coulter, A63880) and eluted in 40 μL of water.

PCR products ran as a smear on agarose gel. The smears were specific, because they failed to appear when the mESCs were not transfected.

### Genomic DNA preparation for linear amplification and UMI-PCR

To eliminate I-SceI sites that were not cut during DSB induction and limit the size of PCR extension, 2 μg genomic DNA from mESCs (both, with DSB induction and control without the induction) was digested using 1 μl 100 U/μL XbaI (NEB, #R0145T) and 4 μl 5 U/μL I-SceI (NEB, #R0694L) in a 50-μL final volume for 3 h at 37 °C. Digested DNA was column-purified (QIAGEN, QIAquick PCR purification kit #28104).

### UMI-LA (linear amplification)

A 500 ng of genomic DNA obtained after I-SceI/XbaI digestion was used as a template in 50 μL of Q5 DNA polymerase reaction mix (NEB, M0491S), using 50 nM of a UMI-containing primer U1 in the following cycling conditions: 98 °C for 4 min; 98°C for 30s, 60 °C for 1 min, and 72 °C for 1 min (50 cycles); and 72 °C for 5 min. Products of linear amplification were purified using Agencourt AM Pure XP beads (Beckman Coulter, A63880) and eluted in 40 μL of water. Linear amplification was repeated 4 times for every sample to account for the technical variability.

### UMI-PCR

A 500 ng of genomic DNA obtained after I-SceI/XbaI digestion was used as a template in a 50 μL of Q5 DNA polymerase reaction mix (NEB, M0491S), using 50 nM of primers U1 and U2 in the following cycling conditions: 98 °C for 1 min; 98 °C for 20s, 60 °C for 1 min, and 72 °C for 4 min (6 cycles); and 72 °C for 10 min. Products of UMI-PCR were purified using Agencourt AM Pure XP beads (Beckman Coulter, A63880) and eluted in 40 μL of water. UMI-PCR was repeated 4 times for every sample to account for the technical variability.

### Sequencing sample preparation of UMI-amplicons

Products of linear amplification and UMI-PCR were used as a template for indexing PCR. A 50 μL of Q5 DNA polymerase reaction mix (NEB, M0491S) was used. Every sample was amplified using 100 nM of primer U2 (annealing to the Illumina PE1.0 primer) and one of indexing primers with Illumina PE2.0 sequence (UIND1-12). The cycling conditions were as follows: 98 °C for 1 min; 98 °C for 20s, 60 °C for 30s, 72 °C for 4 min (30 cycles), and 72°C for 10 min. Products of the indexing PCR were pooled into 3 final samples: (1) control (without DSB induction), (2) 24 h after I-SceI transfection, and (3) 48 h after I-SceI transfection. The samples were purified using 2% E-Gel EX precast agarose gels (Thermo Fisher Scientific, G401002). Each sample was visualized on a Bio-analyzer (Agilent Technologies) and quantified by qPCR using a Kapa Library Quantification Kit (Kapa Biosystems, KK4835).

### High throughput sequencing

Final samples for both inverse PCR and UMI-amplicons (concentration 4 nM) were sequenced as paired-end reads on HiSeq2500 and NextSeq500 sequencers (Illumina).

### Processing inverse PCR reads

The paired-end reads were pre-processed by custom Python scripts. The forward read consists of the barcode, a fixed 20 nucleotide watermark sequence used for identification, the NlaIII restriction site, and some mouse genome sequence ligated to the NlaIII site. The reverse read consists of the last 25 nucleotides of pT2 and the mouse genome sequence at the insertion site of the transposon. We used seeq version 1.1 (https://github.com/ezorita/seeq) allowing up to three errors (mismatches and indels) to identify the watermark and isolate the barcode sequence. Reads were discarded if the watermark was not found or if the barcode was not between 14 and 24 nucleotides long. We removed the first 25 nucleotides of the reverse read and cut the sequence at the first NlaIII site, if any. The reads for which this sequence was shorter than 20 nucleotides were discarded. The genomic sequences thus obtained were mapped in the mouse genome release mm9 using the GEM mapper build 1.376 [51] with options "-q ignore --unique-mapping". The reason for not using the more recent mouse release mm10 was that the chromatin from Juan et al. [34] were only available in release mm9.

After mapping, the barcodes were clustered using Starcode [52] allowing up to two errors (options "-d2 --print-clusters"). This removes potential sequencing errors and consolidates the barcode sequences.

The barcodes were assigned to a genomic location using custom Python scripts. Unmapped genomic sequences were discarded, and sequences mapping to multiple locations were flagged as such. For each barcode, we collected all the insertion sites that totalled at least 10% of the reads and we computed their diameter, equal to the maximum of their pairwise genomic distances (infinite for two sites on different chromosomes or if one of them maps to multiple locations). If the diameter was greater than 30 nucleotides, the barcode was discarded for being used in reporters mapping to different locations. Otherwise, the barcode was kept and its location was attributed to the most frequent insertion site (they are usually within 1–2 nucleotides of each other because of small mapping artifacts).

### Processing UMI-amplicon reads

Paired-end reads were preprocessed using custom Python scripts. The forward read consists of the barcode, the watermark sequence, and the right half of the second F segment in the orientation of Fig. 1b. The reverse read consists of the UMI and the left half of the first F segment in the orientation of Fig. 1b. Both reads extend the midpoint of the F segment by three nucleotides. If the reporter is uncut or repaired by NHEJ, forward and reverse reads do not overlap. If the reporter is repaired by SSA, forward and reverse reads overlap because there is only one F segment. We can thus isolate the reads from reporters that went through SSA by ensuring that the nucleotides in the mismatch position are reverse-complements of each other in the forward and reverse reads.

Thus, we used seeq with up to 10 errors to identify the half F segments and isolate the nucleotides in mismatched position on the forward and reverse reads, together with the barcode and the UMI. Barcodes and UMIs were clustered with Starcode allowing up to 2 errors (options "-d2 --print-clusters"), and the repair events were quantified for each barcode. The barcode–UMI pairs with only one read per run were discarded. After this operation, UMIs that were associated with more than one barcode were discarded. The remaining UMIs were classified as NHEJ or SSA as explained above, and those classified as SSA were further split into A/T or G/C. This provided for each barcode the full list of events reported by UMIs.

Barcodes with more UMIs reporting NHEJ than SSA and those with only one UMI were removed. Barcodes that passed all these criteria in the control experiments without I-SceI induction were removed. The global proportion of remaining UMIs reporting A/T versus G/C was used as a measure of repair bias.

### Mutual information

The operative definition of mutual information is the Kullback-Leibler between the joint distribution of two variables and their product distribution (whereby we assume independence). Joint and product distributions are particularly easy to compute for categorical variables, which makes mutual information more adapted than the Pearson and Spearman coefficients of correlation in this context.

We collected the barcodes with at least 5 UMIs that appeared in at least two replicates and we assigned them to a single repair outcome by majority vote (i.e., each barcode was called either A/T or G/C, even in case of conflicts between UMIs). For every pair of

replicates where the barcode appears, there are thus 4 possible outcomes. We used the number of occurrences of the 4 outcomes as an estimate of their joint distribution and the product of their margins as an estimate of the product distribution.

We computed the mutual information using the log2 function instead of the natural logarithm so that the result is expressed in bits. There is in general no upper bound on mutual information, but for two categorical variables with two outcomes each, the maximum is 1 bit.

### Neural network training

The networks have three hidden layers with 100 neurons each, and one output layer with 2 neurons. The vectors in output of each hidden layer are batch-normalized and the ReLU activation function is applied (i.e., negative values are set to 0). Batch normalization was shown to improve training speed [30], and the rectified linear unit or ReLU is a common activation function that was empirically shown to mitigate the problem of vanishing gradients [31]. The fist two hidden layers include a Dropout step where a random set of input values are set to 0 with probability 0.3. This was shown to mitigate overfitting by forcing some redundancy in the encoding of the information [32]. Including a Dropout step at the output of the last hidden layer showed no benefit. Finally, the networks have a residual connection bypassing the last two hidden layers. Such residual connections were shown to improve training speed and model performance [29].

The input layer of the null model has 7 neurons (4 for the construct, 1 for the time point, one for the amplification technology, and 1 for the TRIP pool). The input layer of the full model has 87 neurons (the 7 neurons of the null model plus 78 for the chromatin features from Juan et al. [34], plus 1 for the GC-content within 10 kb, plus one for the GC-content within 1 Mb). When predicting the GC-content within 10 kb, the input layer of the full model has 85 neurons (the two variables for the GC content are removed).

The 2 neurons in the output layer encode the two parameters of a Beta distribution. The networks thus transform their input into a distribution over the interval (0,1) that reflects the confidence of the model for all the possible values of the bias at the given genomic location. For instance, when the parameters are both close to 1, the distribution is near uniform and the model is "clueless" about the bias because all the values are equally likely. When one parameter is substantially larger than the other, the model expresses confidence that the bias is strong in the given direction. In line with this interpretation, the loss function that the optimizer sets to minimize is defined as the (negative) log-likelihood of the observations under the local Beta distribution associated with a genomic location.

The datasets were split randomly so as to keep 10% of the records for testing. The remaining 90% were used for training for 100 epochs. At the end of every epoch, the performance of the model was evaluated on the test set.

The neural networks were implemented using custom scripts written in Python 3.7.6 with Pytorch version 1.9.0 for CUDA version 10.2 and Numpy version 1.19.2. Networks were optimized in mini batches of size 256 with the Adam optimizer [53] with a learning rate equal to 0.001.

## UMI-LA on methylated oligonucleotides

We used the oligonucleotides METH-Y and METH-N (Table 2) to measure the rate of spontaneous deamination of cytosines during linear amplification. The oligonucleotides have the same structure, with the sequence GAT<u>C</u>GATC flanked by 12 random nucleotides on either side, where the underlined C is methylated in METH-Y but not in METH-N. The 24 random nucleotides form a bipartite barcode that uniquely identifies the molecule that is used as template during linear amplification; the constant sequence provides a way to estimate the frequencies of errors on all four nucleotides.

UMI-LA was performed with primer U1 in the conditions described above, with approximately 100,000 molecules of either METH-Y or METH-N, each in duplicate. The products were further amplified by PCR as detailed above and were sequenced on an Illumina HiSeq 2000. We also performed UMI-PCR with different times of initial denaturation, from which we concluded that the amount of heating in the protocol shown above produces insufficient amount of spontaneous deaminations (detecting robust differences between methylated and non-methylated CpGs required at least 15 min of initial denaturation, data not shown). For this reason, we did not use the available results of UMI-PCR to infer the methylation status of the reporters.

The reverse reads were clustered with Starcode allowing up to 2 errors. The annealing sequence of the U1 primer TGCAACGAATTCATTAGTGCG was removed using seeq allowing for two errors. Finally, UMIs with fewer than two reads were discarded. A barcode was considered to provide evidence for the error X>Y (e.g., A>G) if it had at least one UMI associated with GATCGATC and at least one UMI associated with said error (e.g., GGTCGATC or GATCGGTC).

To find C>T transitions in F segments, a consensus sequence was built for each UMI where each nucleotide was determined by majority vote among the reads for said UMI. A reporter was considered to have a C>T transition if it had at least one UMI with the expected sequence of the F segment and at least one UMI with a C>T transition.

## Code and data

The data has been deposited in the Gene Expression Omnibus database, under accession identifier GSE141211 [54]. The scripts used in this study are available on Github at https://github.com/cellcomplexitylab/strand_asymmetry under the MIT license [55]. A Docker image with the software to reproduce the results is available on Dockehub at https://hub.docker.com/r/gui11aume/strand_asymmetry. All statistical tests were performed with R version 3.6.3 using default parameters and threshold for statistical significance equal to 0.01.

## Supplementary Information

---

**Additional file 1.** Review history.

---

### Acknowledgements

We would like to thank William R. Engels, Carlos Flores, and Laurent Duret for early discussions about this project and Zsuzsanna Izsvák for kindly providing the Sleeping Beauty constructs. We thank the flow cytometry and the sequencing facility of the CRG for their support. We thank the anonymous reviewers for their constructive comments. Our most

sincere gratitude goes to Fyodor Kondrashov, Maximilian Jösch, and Adam Mott for providing material support during the COVID-19 epidemic.

### Review history
The review history is available as Additional file 1.

### Peer review information

### Authors' contributions
VOP, ARD, LE, ATP, and GJF carried out the experiments. GJF performed the analyses. VOP and GJF wrote the manuscript. The authors read and approved the final manuscript.

### Author's information
Twitter handle: @thegrandlocus (Guillaume J. Filion)

### Funding
We acknowledge the financial support of the Natural Sciences and Engineering Research Council of Canada (NSERC RGPIN-2020-06377), the Spanish Ministry of Economy, Industry and Competitiveness ("Centro de Excelencia Severo Ochoa 2013-2017", Plan Estatal PGC2018-099807-B-I00), of the CERCA Programme/Generalitat de Catalunya, and of the European Research Council (Synergy Grant 609989). VOP was supported by the European Union's Horizon 2020 research and innovation program under the Marie Skłodowska-Curie programme (665385). We also acknowledge the support of the Spanish Ministry of Economy and Competitiveness (MEIC) to the EMBL partnership.

### Availability of data and materials
The raw data has been deposited in the Gene Expression Omnibus database, under accession identifier GSE141211 [54]. The scripts used in this study are available on Github at https://github.com/cellcomplexitylab/strand_asymmetry under the MIT license [55]. A Docker image with the software to reproduce the results is available on Dockehub at https://hub.docker.com/r/gui11aume/strand_asymmetry. Mouse E14 ES cells are available from the American Type Culture Collection under ATCC accession CRL-1821. The cell line has not been authenticated.

## Declarations

### Ethics approval and consent to participate
Not applicable.

### Consent for publication
Not applicable.

### Competing interests
The authors declare that they have no competing interests.

### Author details
[1]Center for Genomic Regulation (CRG), The Barcelona Institute of Science and Technology, Dr. Aiguader 88, 08003 Barcelona, Spain. [2]Present Address: Institute of Science and Technology Austria, Am Campus 1, Klosterneuburg, Austria. [3]Present Address: H12O-CNIO Lung Cancer Clinical Research Unit, i + 12 Research Institute, Spanish National Cancer Research Center (CNIO), Madrid, Spain. [4]University Pompeu Fabra (UPF), Barcelona, Spain. [5]Present Address: Department Biological Sciences, University of Toronto Scarborough, Toronto, Canada.

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

## 
