## [**Additional file 1.** Review history. · Genome Biology]

Review History

First round of review

Reviewer 1

Are you able to assess all statistics in the manuscript, including the appropriateness of statistical tests used? No, I do not feel adequately qualified to assess the statistics.

Comments to author:

Historically, measuring mismatch repair in model organisms has relied on using reporter genes, which by their nature can only measure MMR in a limited number of sequence contexts. Genome-wide techniques have become possible, but they lack the ability to place defined mismatches in specific locations. The current manuscript uses a novel method to place a reporter construct, containing an SceI site flanked on either side by two almost identical direct repeats (they differ by a single nucleotide, "Top" and "Bottom"), at numerous genomic locations whose precise positions can then be mapped unambiguously. They use this system to drive SceI-mediated double strand breaks followed by repair to measure the outcome of mutagenic SSA mapped to particular sites in the genome. They find a strong and reproducible bias towards repair using the Top allele regardless of the nature of the mispair or genomic location.

While the assay system is novel and well designed, there are several issues regarding the interpretation of the results that leave some concern.

Major Concerns

A major concern is that this assay might not actually measure mismatch repair at all during SSA. There is a very clear and reproducible bias towards the "top" allele in the reporter system. These results would run counter to a long literature describing the influence of the nature of the mismatch on MMR activity, both biochemically and in model organisms. Given the data presented it would be inaccurate to state, as done so in the title, that the strand asymmetry influences MMR. They could in fact be reporting on other bias not directly related to mismatch repair. One possibility is intrinsic bias specific to the reporter sequence. Another possibility that the authors discuss is bias in flap processing. This elegant system does have rather clear and directly testable paths to clear this up, however. One such possibility is that in the absence of MMR (e.g. CRISPR inactivation of Msh2) this bias should be entirely eliminated, with all reporters reverting to proportional outcomes closer to 0.5 regardless of the reporter.

There is a 20-fold variability in the number of mapped repair events as a proportion of all unambiguously mapped barcodes. Are these numbers from single transposition/repair events for each construct? Or averages from several transposition/repair replicates? What explains the high variability given the uniform nature of the repair bias?

Minor Concerns

It is not clear how many technical replicates were performed for each reporter construct.

Table 1. C:A construct. Repair Events contains an extra digit.

"All the reporters have similar strand biases" section. Line 59. "...idendetical." Should be "...identical."

Reviewer 2

Are you able to assess all statistics in the manuscript, including the appropriateness of statistical tests used? No, I do not feel adequately qualified to assess the statistics.

Comments to author:

In the current manuscript, Pokusaeva and colleagues have set to investigate nucleotide preferences in the mismatch repair system and to address the influence of the chromatin context in the output of MMR. For this, they made use of the TRIP (Thousands of Reporters Integrated in Parallel) assay, which was specifically adapted to score for nucleotide biases during MMR. Doing so, they identify features which can indeed bias the outcome of MMR, in the form of a strand rather than nucleotide bias. They further discuss in their conclusion that this strand bias observed in their assay may reflect the influence of the size of the flaps in their construct. Using machine learning approaches, they also provide evidence that the chromatin context in which MMR operates has little, if any, influence on the output of MMR.

The question is definitely of importance to the field of genome stability. It has implications in understanding the mutational landscape in disease such as cancer, for which extensive characterization is still ongoing.

The approach is sound, in the form of very controlled experiment performed in a variety of contexts at the same time and is definitely innovative in the context of MMR. The provided data provided are highly concordant and reproducible, which is nicely exemplified by the strand swap construct. There are, like always, some limitations originating from the design (the mismatch produced in the TRIP construct may not be directly comparable to mismatches introduced during replication) but these points are discussed and the conclusions appear solid. However, in the present state, the study seems somehow a little bit underdeveloped. The reason for the strand bias observed in this specific system is only speculated and the interpretation that the chromatin context has no effect on the MMR output is still preliminary (yet put forward in the manuscript). While we realize that addressing the strand bias may require extensive work and may be more suitable for a futur manuscript, the authors should be able to expand experimental evidence regarding the impact of chromatin (see main point 1). Without this we would recommend to either entirely remove the chromatin part of the manuscript or to significantly tune down the conclusion.

Major Point 1:

There may be a limitation in the interpretation that the chromatin context has little effect on the MMR output.

First, the flanking transposon sequences could play an insulating role towards the immediate chromatin context, which will then not efficiently "invade" the reporter construct. This would cancel out at least some (if not all) potential effects from the chromatin state at the site of lesion. The authors should challenge the system, for instance deleting histone modifying enzymes or

treating with drugs (TSA...). Without a direct experimental evidence showing that such a treatment does not affect MMR output (or else showing that the reporter-carrying transposon indeed display the chromatin state where it is integrated) it is difficult to conclude that the chromatin state has no effect on MMR.

Second, the contribution of the chromatin state might have been hidden by the original design. The same experiment with a symmetric construct (equivalent flap), where there might be no strand bias anymore, may allow to uncover the effect of chromatin. This should be discussed.

Major Point 2:

There are several points that are not immediately clear regarding the interpretation of machine learning results, at least for non-expert in this field.

First, the methods are very minimally described for this part both in the text and methods section. For example, the text reports that 10% of observations "were kept for testing (see Materials and Methods)", but none of this is actually mentioned in the methods.

It is also very difficult to understand how the entire dataset linking the repair outcome with the features reporting the chromatin context was constructed, specifically which number was actually computed to characterize a given feature (eg, the raw count over a fixed window around the reporter insertion site, ...). This can be important when assessing histone modifications since some can give very strong local enrichment, while others tend to spread more loosely over broad domains. The clarity of the manuscript would also be improved by explaining with more details the interpretation of the provided learning curves.

Also, it is very difficult to understand how the null model in Figure 5B (without chromatin) was trained ("this model can only learn the genome-wide averages per experiments").

All these points are worth clarifying for readers.

Minor points:

Table 1 shows that the number of mapped reporters and/or repair events can vary quite a lot from one experiment to the next (614 to >26000 mapped repair events for example). It is not obvious if these differences directly come from the construct, individual experiments or any other reasons. While this by no means invalidate the authors' conclusions, it is worth discussing this and the potential impact on the results a little bit more. This would help to understand the subtleties, and maybe potential limitations of the TRIP assay and should be beneficial for people willing to adopt such methods, especially in the field of genome stability.

Reviewer 3

Are you able to assess all statistics in the manuscript, including the appropriateness of statistical tests used? Yes, and I have assessed the statistics in my report.

Comments to author:

The manuscript by Pokusaeva et al. examines mismatch repair within the natural genomic environment through the use of an integrated reporter construct throughout the genome of ES cells. The reporter assay utilized is a very elegant and systematic approach. The manuscript is very well written and the results are thoroughly vetted and convincing. The primary conclusions

from the study is that there is a prevalent strand bias in the mismatch repair machinery and that chromatin structure does not impact mismatch repair outcomes.

The results are interesting, yet the investigations into the impact of chromatin on mismatch repair could be improved by providing more information as to the chromatin of the integrated reporters.

It is not clear that the reporters adopt the chromatin organization of the surrounding loci.

Furthermore, it is not clear if the integration sites provide a comprehensive view of the chromatin architecture throughout the genome. Some sort of direct comparisons with chromatin states and/or histone modifications with the integration sites and genome averages would be useful.

Overall, while the results are compelling, I question whether the significance is appropriate for Genome Biology. I believe the significance of the findings could be improved if more mechanistic insight was provided for the basis of the strand bias. In addition, if links could be made to a bias in cancers with functional and dysfunctional mismatch repair, significance could also be improved. Regardless, I believe this is a very well executed study with interesting results.

Small point: The sentences on lines 53-55 are a bit confusing. Is the number of integrations from 9-50 thousand referring to different experiments?

PREAMBLE

We are pleased to submit a revised version of the manuscript. However, this proved difficult because of the ongoing COVID-19 epidemic. A few weeks after receiving the response to our initial submission, we had to close the laboratory at the CRG in the midst of the first wave of COVID-19. The work contract of the principal investigator expired by the end of the lockdown, so the laboratory closed permanently before we had time to start working on a revised version. In the year that followed, we were able to carry out some experiments thanks to the generosity of our colleagues in Austria, Spain and Canada. However, some experiments were unsuccessful and we are not able to repeat them for lack of personnel, funding and material.

In spite of these difficulties, we think that the present version is worth submitting. It shows some concrete improvement over the previous submission thanks to the success of a key experiment. We apologize to the Reviewers for not being able to honor their request. We hope that they will nevertheless appreciate the scientific merit of the present work.

SUMMARY OF MAJOR CHANGES

- 1) The last section of the Results “Asymmetries between flaps influence the repair bias” and the new Figure 6 present experimental evidence for the model introduced in the Discussion of the previous version.
- 2) The mismatch label as C:A was actually G:T. The labelling error has been corrected throughout the manuscript.
- 3) The total number of events has increased (Table 1). Several barcodes had been excluded due to a mistake in the early design of the Docker container to reproduce our results. This does not affect any of the previous conclusions.
- 4) The number of chromatin features used for machine learning has gone down (Figure 5). We had added some extra features to the set of Juan *et al.* (reference 29 in the text) but with a resolution of 5 kb. To keep the original resolution of 200 bp, we had to remove the extra features. This does not affect any of the previous conclusions.

=====

Reviewer #1: Historically, measuring mismatch repair in model organisms has relied on using reporter genes, which by their nature can only measure MMR in a limited number of sequence contexts. Genome-wide techniques have become possible, but they lack the ability to place defined mismatches in specific locations. The current manuscript uses a novel method to place a reporter construct, containing an SclI site flanked on either side by two almost identical direct repeats (they differ by a single nucleotide, "Top" and "Bottom"), at numerous genomic locations whose precise positions can then be mapped unambiguously. They use this system to drive SclI-mediated double strand breaks followed by repair to measure the outcome of mutagenic SSA mapped to particular sites in the genome. They find a strong and reproducible bias towards repair using the Top allele regardless of the nature of the mispair or genomic location.

While the assay system is novel and well designed, there are several issues regarding the interpretation of the results that leave some concern.

Major Concerns

A major concern is that this assay might not actually measure mismatch repair at all during SSA. There is a very clear and reproducible bias towards the "top" allele in the reporter system. These results would run counter to a long literature describing the influence of the nature of the mismatch on MMR activity, both biochemically and in model organisms. Given the data presented it would be inaccurate to state, as done so in the title, that the strand asymmetry influences MMR. They could in fact be reporting on other bias not directly related to mismatch repair. One possibility is intrinsic bias specific to the reporter sequence. Another possibility that the authors discuss is bias in flap processing. This elegant system does have rather clear and directly testable paths to clear this up, however. One such possibility is that in the absence of MMR (e.g. CRISPR inactivation of Msh2) this bias should be entirely eliminated, with all reporters reverting to proportional outcomes closer to 0.5 regardless of the reporter.

Absolutely. We have intended to carry out the assay upon MSH2 or MSH6 knock-downs, but unfortunately the sequencing runs failed and we were not able to repeat the experiment (see preamble). However, we bring experimental evidence that the bias is due to flap processing (see Figure 6). We have not formally tested that the MMR pathway is involved, but we consider that it is beyond doubt that the biases occur during the processing of the mismatch by the DNA repair system. Since we are not able to resolve this question entirely, we propose to change the title to "*Strand asymmetry influences mismatch resolution during single-strand annealing*".

There is a 20-fold variability in the number of mapped repair events as a proportion of all unambiguously mapped barcodes. Are these numbers from single transposition/repair events for each construct? Or averages from several transposition/repair replicates? What explains the high variability given the uniform nature of the repair bias?

Reviewer #2 also raises a similar issue.

One repair event corresponds to the quantification of one barcode in one UMI-LA / UMI-PCR. The UMIs give us confidence that those are indeed independent events. If the barcode is mapped to a known location, the event is counted as a mapped event. We have clarified the meaning of the numbers in the legend of the table.

The experiments presented in the manuscript started in 2011; they were performed by different people in different places with different equipment, so it has been difficult to obtain consistent results at every step. The main issue is the T:C mismatch, for which there are much fewer events. Those assays were performed simultaneously with the A:G and A:C mismatches, which led to contaminations between the experiments. Each construct contains a watermark (a set of unique nucleotides) allowing us to identify contaminating reads and to remove them automatically. Performed last in this batch, the T:C experiment lost more reads to contaminations, meaning that many barcodes did not meet the quality standard of more than one read per UMI. We now mention the large spread of the numbers in Table 1 when describing it in the main text.

Note: The answer above was replicated to answer the last (minor) point of Reviewer #2.

Minor Concerns

It is not clear how many technical replicates were performed for each reporter construct.

Each UMI-PCR / UMI-LA was performed as 4 replicates per time point and per construct. There are a total of 8 UMI-PCR + 8 UMI-LA per construct. Two UMI-PCRs for the A:G mismatch at 48 hours failed to give any reads. For every construct we made two TRIP pools; every replicate (UMI-PCR or UMI-LA) was performed on both TRIP pools simultaneously. This information has been added to the text.

Table 1. C:A construct. Repair Events contains an extra digit.

As explained in point 3) of the summary of major changes, the numbers in the table have changed.

"All the reporters have similar strand biases" section. Line 59. "...idendental." Should be "...identical."

The typo has been fixed.

=====

Reviewer #2: In the current manuscript, Pokusaeva and colleagues have set to investigate nucleotide preferences in the mismatch repair system and to address the influence of the chromatin context in the output of MMR. For this, they made use of the TRIP (Thousands of Reporters Integrated in Parallel) assay, which was specifically adapted to score for nucleotide biases during MMR. Doing so, they identify features which can indeed bias the outcome of MMR, in the form of a strand rather than nucleotide bias. They further discuss in their conclusion that this strand bias observed in their assay may reflect the influence of the size of the flaps in their construct. Using machine learning approaches, they also provide evidence that the chromatin context in which MMR operates has little, if any, influence on the output of MMR.

The question is definitely of importance to the field of genome stability. It has implications in understanding the mutational landscape in disease such as cancer, for which extensive characterization is still ongoing.

The approach is sound, in the form of very controlled experiment performed in a variety of contexts at the same time and is definitely innovative in the context of MMR. The provided data provided are highly concordant and reproducible, which is nicely exemplified by the strand swap construct. There are, like always, some limitations originating from the design (the mismatch produced in the TRIP construct may not be directly comparable to mismatches introduced during replication) but these points are discussed and the conclusions appear solid. However, in the present state, the study seems somehow a little bit underdeveloped. The reason for the strand bias observed in this specific system is only speculated and the interpretation that the

chromatin context has no effect on the MMR output is still preliminary (yet put forward in the manuscript). While we realize that addressing the strand bias may require extensive work and may be more suitable for a future manuscript, the authors should be able to expand experimental evidence regarding the impact of chromatin (see main point 1). Without this we would recommend to either entirely remove the chromatin part of the manuscript or to significantly tune down the conclusion.

We thank the Reviewer for this summary of our manuscript. In spite of technical difficulties linked with the COVID-19 epidemic (see preamble), we were able to bring experimental evidence that the bias is due to flap processing, regardless of the insertion site of the reporter (Figure 6). Our answers below will clarify this statement.

Major Point 1:

There may be a limitation in the interpretation that the chromatin context has little effect on the MMR output.

First, the flanking transposon sequences could play an insulating role towards the immediate chromatin context, which will then not efficiently "invade" the reporter construct. This would cancel out at least some (if not all) potential effects from the chromatin state at the site of lesion. The authors should challenge the system, for instance deleting histone modifying enzymes or treating with drugs (TSA...). Without a direct experimental evidence showing that such a treatment does not affect MMR output (or else showing that the reporter-carrying transposon indeed display the chromatin state where it is integrated) it is difficult to conclude that the chromatin state has no effect on MMR.

Reviewer #3 also raises a similar issue.

We agree with this point. We have tried to perform the repair assay after treating the cells with TSA. Even though we got visual confirmation that the treatment was effective, it turned out that the quality of the sequencing samples was very low and we did not have enough reads to conclude anything from these experiments.

The pT2 backbone is sensitive to transcriptional position effects in those cells (our unpublished results) and in *Drosophila* we could show that it acquires the chromatin features of its surroundings (Corrales et al., 2017, PMID:28420691). This backbone is also used by other researchers for reporter assays; for instance François Spitz used it for fine-scale mapping of enhancer activities (e.g., Chen et al. 2013, PMID: PMC3618316). The repeated fragment is a subsequence from the GFP, also widely used for reporter assays. In the absence of direct experimental evidence, we cannot formally rule out that the reporter acts as an insulator for the neighboring chromatin, but there are indications it does not contain promoters, enhancers or insulators.

We mention this limitation in the Discussion.

Note: The answer above was replicated to answer the first point of Reviewer #3.

Second, the contribution of the chromatin state might have been hidden by the original design. The same experiment with a symmetric construct (equivalent flap), where there might be no strand bias anymore, may allow to uncover the effect of chromatin. This should be discussed.

We have performed a related experiment, whereby we used CRISPR to cut different sites of the construct. Using a guide RNA that cuts at approximately the same site as I-SceI, we obtained the same results as before, but when we used a guide RNA that cuts at the opposite side of the spacer between the repeats, we reversed the bias completely (Figure 6).

This is strong evidence that the bias is due to the asymmetry of flap processing. It is important to highlight that this practically rules out a role for chromatin in this assay. Indeed, the bias is reversed on the very same reporter, inserted at the same location, *i.e.* in the same chromatin context of the genome. The bias reversal appeared to be the same throughout the genome (Figure 6).

Still, the possibility remains that the bias reversal is observed only because a putative shielding effect of the reporter prevents the chromatin context from exerting an effect.

We mention this in the Discussion, together with the limitation mentioned in the previous point.

Major Point 2:

There are several points that are not immediately clear regarding the interpretation of machine learning results, at least for non-expert in this field.

First, the methods are very minimally described for this part both in the text and methods section. For example, the text reports that 10% of observations "were kept for testing (see Materials and Methods)", but none of this is actually mentioned in the methods.

It is also very difficult to understand how the entire dataset linking the repair outcome with the features reporting the chromatin context was constructed, specifically which number was actually computed to characterize a given feature (eg, the raw count over a fixed window around the reporter insertion site,). This can be important when assessing histone modifications since some can give very strong local enrichment, while others tend to spread more loosely over broad domains. The clarity of the manuscript would also be improved by explaining with more details the interpretation of the provided learning curves.

Also, it is very difficult to understand how the null model in Figure 5B (without chromatin) was trained ("this model can only learn the genome-wide averages per experiments").

All these points are worth clarifying for readers.

Thank you for bringing up this weakness in the text. We have added the details in the Methods section. We did our best to clarify the analyses related to this part.

Minor points:

Table 1 shows that the number of mapped reporters and/or repair events can vary quite a lot from one experiment to the next (614 to >26000 mapped repair events for example). It is not obvious if these differences directly come from the construct, individual experiments or any other reasons. While this by no means invalidate the authors' conclusions, it is worth discussing this and the potential impact on the results a little bit more. This would help to understand the subtleties, and maybe potential limitations of the TRIP assay and should be beneficial for people willing to adopt such methods, especially in the field of genome stability.

Reviewer #1 also raised a similar issue.

The experiments presented in the manuscript started in 2011; they were performed by different people in different places with different equipment, so it has been difficult to obtain consistent results at every step. The main issue is the T:C mismatch, for which there are much fewer events. Those assays were performed simultaneously with the A:G and A:C mismatches, which led to contaminations between the experiments. Each construct contains a watermark (a set of unique nucleotides) allowing us to identify reads from cross-contaminations between experiments and to remove them automatically. Performed last in this batch, the T:C experiment lost more reads to contaminations, meaning that many barcodes did not meet the quality standard of more than one read per UMI. We now mention the large spread of the numbers in Table 1 when describing it in the main text.

Note: The answer above was replicated to answer the second point of Reviewer #1.

=====

Reviewer #3: The manuscript by Pokusaeva et al. examines mismatch repair within the natural genomic environment through the use of an integrated reporter construct throughout the genome of ES cells. The reporter assay utilized is a very elegant and systematic approach. The manuscript is very well written and the results are thoroughly vetted and convincing. The primary conclusions from the study is that there is a prevalent strand bias in the mismatch repair machinery and that chromatin structure does not impact mismatch repair outcomes.

The results are interesting, yet the investigations into the impact of chromatin on mismatch repair could be improved by providing more information as to the chromatin of the integrated reporters. It is not clear that the reporters adopt the chromatin organization of the surrounding loci.

Reviewer #2 raised a similar point.

The pT2 backbone is sensitive to transcriptional position effects in those cells (our unpublished results) and in *Drosophila* we could show that it acquires the chromatin features of its surroundings (Corrales et al., 2017, PMID:28420691). This backbone is also used by other researchers for reporter assays; for instance François Spitz used it for fine-scale mapping of enhancer activities (e.g., Chen et al. 2013, PMID: PMC3618316). The repeated fragment is a subsequence from the GFP, also widely used for reporter assays. In the absence of direct

experimental evidence, we cannot formally rule out that the reporter acts as an insulator for the neighboring chromatin, but there are indications it does not contain promoters, enhancers or insulators.

We mention this limitation in the Discussion.

Note: The answer above was replicated to answer the Major Point 1 of Reviewer #2.

Furthermore, it is not clear if the integration sites provide a comprehensive view of the chromatin architecture throughout the genome. Some sort of direct comparisons with chromatin states and/or histone modifications with the integration sites and genome averages would be useful.

Thank you for the suggestion. We have added a panel to Figure 5 where we show the outcome of the assay in different chromatin contexts.

Overall, while the results are compelling, I question whether the significance is appropriate for Genome Biology. I believe the significance of the findings could be improved if more mechanistic insight was provided for the basis of the strand bias. In addition, if links could be made to a bias in cancers with functional and dysfunctional mismatch repair, significance could also be improved. Regardless, I believe this is a very well executed study with interesting results.

In this version we bring substantial evidence that the bias is caused by the asymmetry of the flaps. We designed guide RNAs to cut the reporter at different sites and obtained a complete reversion of the bias (Figure 6). These results are significant on three accounts:

- 1) They establish the existence of a mechanism that was hitherto unknown.
- 2) They show that other biological covariates of the damage are secondary in this assay.
- 3) They exclude the possibility that the biases were caused by the design of the reporter.

Future studies will figure out the generality of the phenomenon and the identity of the molecular actors. Our opinion is that those results are a landmark in understanding the mechanisms of mutational biases, and as such, their significance is appropriate for the genomics community and the readership of Genome Biology at large.

Small point: The sentences on lines 53-55 are a bit confusing. Is the number of integrations from 9-50 thousand referring to different experiments?

The numbers refer to different constructs, *i.e.*, different types of mismatches. This has been clarified in the text.

Second round of review

Reviewer 2

In their revised manuscript, Pokuaseva et al provide new experimental data highlighting the importance of the position of the DSB relative to the 2 F segments in their TRIP reporter (Figure 6). These data do indeed show that the structure of the intermediate plays a very important role in the observed strand bias following repair by MMR. These data provide substance to the model suggested in the original submission, in which the extent and geometry of the flaps are major contributors to the output of MMR. This effect appears to outweigh the potential effect of the insertion site, which the author interpret as a lack of influence by the surrounding chromatin context.

Reply to Major point 1 :

This lack of influence of the chromatin context was not substantiated by many experimental data but rather inferred from the lack of strong influence of the insertion site and the output of the machine learning experiments. This made for the strongest criticism of the original manuscript (major point 1 in the original review)

In this new version, the authors provide no new information regarding the ability of the inserted reporters to fully or even partially acquire the chromatin context present at the insertion sites (a point also raised by referee 3), a key feature to support their conclusion of a lack of influence of the chromatin context. New citations are welcome additions since they indeed suggest that this is likely the case but direct experimental support is lacking.

The authors also provide more analysis of their data, notably by showing that MMR outcome is similar for regions locally enriched in H3K36me3 or H3K9me3 at the insertion site. While this piece of data is indeed in favor of the authors' main hypothesis regarding the lack of strong influence of chromatin features on the observed strand bias, the analysis could have gone a little bit deeper. For example, such comparisons could have been extended to other chromatin features which are considered diagnostic of different chromatin states in mammals such as H3K4me3 (Promoters), H3K27me3 (Polycomb/bivalent domains in ES cells) H3K4me1/H3K27ac (enhancers).

Also, the CRISPR-Cas9 experiments are not used at all to assess a potential impact of the chromatin context and the influence of the insertion site is not addressed in details.

The authors also report a failed experiment attempting to globally challenge chromatin structure, which could have been very informative.

Thus, the authors failed to directly address Major point 1 and to provide strong additional evidence that they can readily interpret the influence of the chromatin context over the position of the DSB in the TRIP reporter. It could therefore be more suitable to conclude that the strand bias does not seem to be strongly influenced by the insertion site, which can be interpreted as a potential lack of effect of the chromatin context but does not, in the current state, provide a definitive answer. This would also mean tuning down the following sentences in the abstract and results :

"These results suggest that the processing of the double-strand break has a major influence on the repair of mismatches during single-strand annealing, irrespective of the surrounding chromatin context"

"Mismatch repair on the reporters is independent of chromatin features"

Reply to major point 2:

The authors made an effort to better explain the details of the machine learning experiments and the interpretation of the results. A few points could also be made even clearer:

For example, how can the null model without chromatin features learn anything in figure 5d and 5f apart from batch effects or technical biases? While this doesn't impact the conclusions, it could be interesting to discuss in a few sentences. Similarly, how could the same null model be able to partially predict the local G/C content (albeit with a lower efficiency compared to the full model) if the insertion site does not have a strong role in the observed strand bias?

Reply to minor point 1, also raised by referee 1:

The reply is satisfactory

Altogether, the authors have not been able to strengthen their conclusion regarding the contribution of chromatin features on MMR. The new data provides a new interesting result regarding a potentially very important role for the processing of flaps for strand discrimination in MMR but fail to provide any new mechanistic insights.

Yet, it appears clearly that it will be very difficult to perturbation or loss-of function experiments to comprehensively address these points. The experimental strategy is original, the strand bias result is very strong and the new data suggesting a key role for the flaps in repairing the mismatches is new.

This study could be considered for publication in *Genome Biology*, provided that the chromatin component is either strengthened (see some suggestions above) or even more drastically tuned down in the text.

Reviewer 3

The revised manuscript now has additional data supporting the strand bias in mismatch repair. In addition, they partially address concerns raised regarding the chromatin context of the reporter system by comparing integration sites in silent genes, expressed genes and intergenic regions. While this is informative it falls short of demonstrating the role, or lack thereof, of chromatin in influencing mismatch resolution. Given the vast amount of attention and investigations into the influence of chromatin modifications on different repair mechanisms, I believe this information is needed in order to substantiate their results and extend them to endogenous mismatch repair mechanisms.

In addition, as stated in the original review, the significance of the study could be bolstered by an examination of mutations in cancers with dysfunctional mismatch repair.

Overall, the study is intriguing and well-presented, yet needs additional data to support generality of conclusions and significance.

SUMMARY OF THE CHANGES

1. New experimental data

We have added some data suggesting that the chromatin of the integrated reporters is not “static”. In 2018 we noticed that linear amplification gave many UMIs with CG>TG alterations. We reasoned that it must be due to DNA methylation and used a positive control to test it. We dropped this research when we realized that the chances that a methylated CG would give a UMI with TG were less than 1%. Thinking again about it for this manuscript, we reasoned that we cannot know the exact methylation status of every CG, but we have sufficient power to find differences in global trends. Using this indirect assay, we find that reporters are less methylated when they are inserted in promoters (the undergrad student who carried out the experiments in 2018 was added as an author).

2. Update on the effect of chromatin

The first version of the manuscript was written from the perspective of the strand bias. The claims that chromatin had no effect meant that it does not explain the bias. We now realize that the wording is confusing and we tried to describe the results more accurately. While doing so, we reviewed the evidence and realized that some small effects of chromatin are detectable after removing batch effects. We have clarified the purpose of the deep learning model and its limitations.

To facilitate the work of the reviewers, the changes are highlighted in red in the present version of the manuscript.

Reviewer #2: In their revised manuscript, Pokuaseva et al provide new experimental data highlighting the importance of the position of the DSB relative to the 2 F segments in their TRIP reporter (Figure 6). These data do indeed show that the structure of the intermediate plays a very important role in the observed strand bias following repair by MMR. These data provide substance to the model suggested in the original submission, in which the extent and geometry of the flaps are major contributors to the output of MMR. This effect appears to outweigh the potential effect of the insertion site, which the author interpret as a lack of influence by the surrounding chromatin context.

We would like to thank the Reviewer for spending time on our manuscript and sharing his / her expertise. We hope that our comments below address the remaining concerns. If not, we would be grateful if the Reviewer could share his / her concerns, and perhaps give us some plausible alternatives so that we can try to test them for completeness.

Reply to Major point 1 :

This lack of influence of the chromatin context was not substantiated by many experimental data but rather inferred from the lack of strong influence of the insertion site and the output of the

machine learning experiments. This made for the strongest criticism of the original manuscript (major point 1 in the original review)

In this new version, the authors provide no new information regarding the ability of the inserted reporters to fully or even partially acquire the chromatin context present at the insertion sites (a point also raised by referee 3), a key feature to support their conclusion of a lack of influence of the chromatin context. New citations are welcome additions since they indeed suggest that this is likely the case but direct experimental support is lacking.

The authors also provide more analysis of their data, notably by showing that MMR outcome is similar for regions locally enriched in H3K36me3 or H3K9me3 at the insertion site. While this piece of data is indeed in favor of the authors' main hypothesis regarding the lack of strong influence of chromatin features on the observed strand bias, the analysis could have gone a little bit deeper. For example, such comparisons could have been extended to other chromatin features which are considered diagnostic of different chromatin states in mammals such as H3K4me3 (Promoters), H3K27me3 (Polycomb/bivalent domains in ES cells) H3K4me1/H3K27ac (enhancers).

We have modified Figure 5 to show additional histone marks. As mentioned above, we find a small effect of H3K36me3 and H3K79me2 on the bias. The text has been changed accordingly.

Also, the CRISPR-Cas9 experiments are not used at all to assess a potential impact of the chromatin context and the influence of the insertion site is not addressed in details. The authors also report a failed experiment attempting to globally challenge chromatin structure, which could have been very informative.

Thus, the authors failed to directly address Major point 1 and to provide strong additional evidence that they can readily interpret the influence of the chromatin context over the position of the DSB in the TRIP reporter. It could therefore be more suitable to conclude that the strand bias does not seem to be strongly influenced by the insertion site, which can be interpreted as a potential lack of effect of the chromatin context but does not, in the current state, provide a definitive answer. This would also mean tuning down the following sentences in the abstract and results :

"These results suggest that the processing of the double-strand break has a major influence on the repair of mismatches during single-strand annealing, irrespective of the surrounding chromatin context"

"Mismatch repair on the reporters is independent of chromatin features"

As mentioned above, we have reworded the manuscript for greater accuracy. We have also added some experimental data, which are shown in figure 6 and described in section An assay to infer the methylation status of the reporters.

Reply to major point 2:

The authors made an effort to better explain the details of the machine learning experiments and the interpretation of the results. A few points could also be made even clearer: For example, how can the null model without chromatin features learn anything in figure 5d and 5f apart from batch effects or technical biases? While this doesn't impact the conclusions, it could be interesting to discuss in a few sentences.

This is precisely the point: the null model can learn only those “uninteresting” biases and the fact that the full model cannot do better shows how poorly it fits the data. This has been clarified in the text.

Similarly, how could the same null model be able to partially predict the local G/C content (albeit with a lower efficiency compared to the full model) if the insertion site does not have a strong role in the observed strand bias?

This is an excellent question! The null model learns something in the first 2–3 epochs; what is it? The output of our model is not a number but a distribution. In other words, the model provides an uncertainty around the estimate and it needs to fit both the mean and the variance of the target value. The GC-content does not vary so much around the genome-wide average, so the loss decreases because the model learns that it can reduce the variance to make better predictions (even though the mean remains the same).

This is not happening when predicting the bias because the fluctuations are much wider (the score spans the full dynamic range between 0 and 1).

Reply to minor point 1, also raised by referee 1:
The reply is satisfactory

Altogether, the authors have not been able to strengthen their conclusion regarding the contribution of chromatin features on MMR. The new data provides a new interesting result regarding a potentially very important role for the processing of flaps for strand discrimination in MMR but fail to provide any new mechanistic insights.

Yet, it appears clearly that it will be very difficult to perturbation or loss-of function experiments to comprehensively address these points. The experimental strategy is original, the strand bias result is very strong and the new data suggesting a key role for the flaps in repairing the mismatches is new.

This study could be considered for publication in Genome Biology, provided that the chromatin component is either strengthened (see some suggestions above) or even more drastically tuned down in the text.

Reviewer #3: The revised manuscript now has additional data supporting the strand bias in mismatch repair.

In addition, they partially address concerns raised regarding the chromatin context of the reporter system by comparing integration sites in silent genes, expressed genes and intergenic regions. While this is informative it falls short of demonstrating the role, or lack thereof, of chromatin in influencing mismatch resolution. Given the vast amount of attention and investigations into the influence of chromatin modifications on different repair mechanisms, I believe this information is needed in order to substantiate their results and extend them to endogenous mismatch repair mechanisms.

We would like to thank the Reviewer for spending time on our manuscript and sharing his / her expertise. We hope that our comments below address the remaining concerns. If not, we would be grateful if the Reviewer could share his / her concerns, and perhaps give us some plausible alternatives so that we can try to test them for completeness.

In addition, as stated in the original review, the significance of the study could be bolstered by an examination of mutations in cancers with dysfunctional mismatch repair.

We absolutely agree with this suggestion. The reason we did not include this kind of analysis is that we do not see which kind of data would be appropriate. To make any prediction, we need to know something about the primary DNA lesion because it is the source of the bias in our model. We cannot think of a biological sample where this information is available. If we focused on events of recombinations between tandem repeats, we would not know the location of the double-strand break that may have triggered it, so we would not be able to make any prediction about the expected outcome. Likewise, the expectations are unchanged whether mismatch repair is functional or not: either there is no bias because the mismatch is not repaired, or there is no bias because the initial double-strand break can be anywhere.

Actually, we consider that this is a major strength of our work that it would be extremely difficult to discover this bias in sequencing data. The strand-bias does not amount to a mutational signature, but it says something important about how DNA repair works. Eventually, we could even turn the argument on its head and use existing mutational biases during recombination between tandem repeats in order to infer the position of the initiating double-strand break. However, the argument is circular until our model can be validated independently.

If the Reviewer would like to suggest some analyses that can be done to either prove or disprove the proposed mechanism (or some other claims), we would be more than happy to perform them. We are eager to put our interpretations to the test in every constructive way.

Overall, the study is intriguing and well-presented, yet needs additional data to support generality of conclusions and significance.

As mentioned above, we have added some experimental data, which are shown in figure 6 and described in section An assay to infer the methylation status of the reporters.

Third round of review

Reviewer 2

In their revised manuscript, the authors have addressed all the previous points raised in a satisfactory manner. They have added additional analyses regarding the chromatin context showing a slight effect of H3K36me3 and H3K79me2. They also added experimental data that allow to infer the methylation status of the reporter, and showed that reporters integrated in promoters tend to be protected from methylation, which is an evidence that reporters are indeed not isolated from their surrounding chromatin.

Few typos should be corrected:

-Page 12, part "The insertion site has a weak influence on the repair bias"

"(from 0.8% coverage for H2Aub1 and H3K20me4, to 5% coverage for H3K36me3)".

The authors probably mean H4K20me3 rather than H3K20me4

-Page 20, discussion :

"Therefore, it is possible that H3K36me3 and H3K79me3 interfere with the Mbd4 pathway."

The authors probably mean H3K79me2 rather than me3

Authors' response

We thank the Reviewer. The typos have been corrected.